# Unequal gains from remote work during COVID-19 between spouses: Evidence from longitudinal data in Singapore

Zeewan Lee [1]*, Poh Lin Tan[2], Jie-Sheng Tan-Soo[1]

1 Lee Kuan Yew School of Public Policy, National University of Singapore, Singapore, Singapore,
2 Institute of Policy Studies, National University of Singapore, Singapore, Singapore

* zeewan.lee@nus.edu.sg

## Abstract

The COVID-19 outbreak and the rise of remote work may have generated distinct labor market outcomes for workers, depending on their occupation and adaptability to changes in the mode of work. Using panel data of married spouses around the time of the government-mandated lockdown in Singapore and difference-in-differences models, we examine the effects of remote work arrangements on salary income, work hours, and wages. We find that the benefits of remote work during the pandemic were not distributed evenly across male and female spouses—translating into longer and significant gains in salary income only for male workers who adopted full remote work arrangements. In contrast, female remote workers' income gains were limited by disproportionately heavier household responsibilities (i.e., chores), which in turn led to constrained work hours.

## 1. Introduction

The COVID-19 pandemic led to widespread and profound changes to labor markets around the world. As governments enacted unprecedented measures such as lockdowns and restrictions on economic activity to limit the spread of the virus and prevent surges in hospitalizations and mortality rates, industries and businesses faced precipitous declines in demand, while many workers were suddenly required to shift to remote work arrangements. In East Asia and the Pacific, where the virus was first detected, firm sales fell by 38–58% on average in the second quarter of 2020 relative to 2019, forcing businesses to reduce operating capacity, shut temporarily, or go out of business, especially small and medium sized enterprises which had lower access to credit or were slower to adapt e-commerce [1]. Employees, in turn, experienced deteriorating labor market outcomes such as reduced hours, wage cuts, job loss, and logistical challenges of changing work arrangements. The pandemic generated devastating consequences in the labor market, especially for the vulnerable workers earning lower-income, contract-based, or paid by the hour [2–4].

**Data availability statement:** We would like to seek an exemption to the journal's data sharing policy as the public deposition of raw data will breach compliance with the protocol approved by the National University of Singapore's IRB (Ref #: LN-17-048). More specifically, our IRB protocol precludes us from publicly releasing the dataset due to sensitive data collected including dates/frequency of sexual activity and fertility. But we would be happy to share anonymized, aggregated data with masked sexual activity/fertility variables with researchers upon request. Interested parties may contact NUS IT at itcare@nus.edu.sg or https://nusit.nus.edu.sg/itcare/ to request the cleaned data shared in the university's data archive (Nbox).

**Funding:** This project was supported by three distinct Start-up Grants at LKY School of Public Policy, National University of Singapore [Internal]. - A-0003976-00-00: Used to purchase analytical software programs that enabled data analyses and to finance the publication fees. - R-603-000-347-115: Used for data collection (primary source) - R-603-000-237-133: Used for data collection There was no additional external funding received for this study.

**Competing interests:** The authors have declared that no competing interests exist.

The wide-spread adoption of remote work—also known as work-from-home (WFH)—is one of the most symbolic labor market changes brought forth by the pandemic. With the technological capacity to facilitate remote operations already existing in many advanced economies including Singapore, the countries rapidly transitioned to adopt remote work as a new mode of work at the COVID-19 outbreak. By April 2020, 59 countries implemented WFH schemes for non-essential workers [5]. Online job postings in 20 countries between 2019 and 2021 show that the share of advertisements for WFH positions more than tripled, and remote work arrangements continue to stay as a part of the new normal today [6,7]. As for Singapore, the country had one of the highest shares of employed persons working remotely in 2020—with 1,094,900, or approximately 49% out of total employed residents working from home (WFH) [8]. This rate exceeded that of several Western countries with the highest shares of remote work arrangements in the same year: 35.2% in the US [9], and 25% Finland, 23.1% Luxembourg [10]. While there is no publicly available report on the share of remote workers before the pandemic in Singapore, Monetary Authority of Singapore kept track of the share of establishments offering 'flexible work arrangements'—a broader notion that encompasses remote work arrangements—and reported that such establishments jumped from less than 50% in 2014 to over 90% in 2021 [11]. Across gender, 43.6% of men and 55.9% of women worked remotely in 2020 in Singapore [8,12].

A stream of COVID-19 literature has investigated into the effect of WFH on gender (in)equality. Studies have shown that the labor market outcomes of women are disproportionately and negatively affected by the pandemic because they, for instance, comprise a larger share of vulnerable employment than male workers do—i.e., a heavy female concentration in service jobs that could not be performed remotely, exposing more women to health risks or temporary/permanent displacement [2–4,13]. At the same time, other researchers have argued that, despite the immediate hardships of female workers documented in the empirical literature, women may eventually gain more from the rise of remote work induced by the pandemic. Case in point, Alon et al. argue that the rise of remote work and other flexible work arrangements could eventually lead to greater gender equality [14].

To add to the ongoing discourse in the literature, we study the impact of adoption of remote work arrangements on the labor market outcomes using an exogenous shock that is the government-mandated lockdown in the second quarter of 2020 that required all who could switch to remote work to adopt the arrangement. A longitudinal data was collected from our first-hand survey (approved by National University of Singapore's IRB Board, Ref#: LN-17–048) in Singapore between 2018 and 2020. The survey was initially collected to understand sociodemographic characteristics, employment statuses, family composition and characteristics, marriage, fertility, and health of married women and their spouses in Singapore at the baseline (wave 1) in 2018, and was retrofitted in 2020 to include questions regarding the pandemic and changing labor market outcomes. Using the 2018–2020 panel data, we conduct difference-in-differences estimation to evaluate the impact of adopting remote work on hourly wages, monthly salary income, and monthly hours worked *during* and *after*

the government-mandated lockdown. Next, we test whether there exist *gender inequalities* in the effect of remote work adoption on respondents' labor market outcomes. Lastly, we explore potential explanations for these gender differences by exploring the mediating roles of household responsibilities.

Our results indicate that the impact of remote work on labor market outcomes varied substantially by gender in 2020. Male WFH workers experienced a modest income loss during the lockdown period (March–June 2020), relative to their non-WFH counterparts. Although they earned higher hourly wages during this period, the income loss was driven by a significant decline in hours worked. However, this trend reversed in the post-lockdown period (November 2020), when male WFH workers experienced modest income gains compared to non-WFH males—driven by sizeable increases in hourly wages, which compensated for their reduced work hours. Among female workers, those who adopted full WFH arrangements recorded slight income gains during the lockdown, relative to non-WFH females, driven by increases in both hourly wages and work hours. However, in the post-lockdown period, female WFH workers experienced income losses, primarily due to a substantial reduction in work hours that outweighed the gains in hourly wages.

Our paper contributes to the literature in several ways. First, we add to the growing literature on the implications of the adoption of remote work, testing whether the rise of remote work arrangements—following the COVID-19 outbreak—exacerbate gender inequality in the labor market outcomes [14]. Our study also adds to a broader, long-standing body of research examining the implications of the 'increased flexibility of work' such as blurred boundaries between work and life, increased spatial and temporal autonomy over work, and evolving norms and expectations [15–19]. Second, we add to the literature by identifying whether the unequal gains and losses associated with remote work across spouses can be explained by (1) their differential time-use in doing household work (e.g., chores, childcare), and (2) the extent of help they receive from domestic helpers and grandparents. Previous studies consistently show that female spouses rather than male spouses took on a greater share of increased domestic responsibilities during the pandemic [20–24]. Our paper builds on these findings by inspecting the mediating effects of the unequal distribution of household work between spouses on their income trajectories, using household-level data on both spouses' time use on household chores and childcare, household composition, and presence of helpers.

Lastly, we expand the geographic scope of the COVID-19 and WFH literature by providing empirical evidence from Singapore. The current literature on the labor market impact of COVID-19 primarily relies on evidence from Western or larger economies. Such evidence may not apply to non-Western smaller economies given the international heterogeneity in the institutional responses to COVID-19, extents of adoption of remote work, labor market structures (i.e., occupational- and industry- distribution of labor), and sociocultural contexts. Indeed, the literature suggests a significant East-West divide in the fight against COVID-19. While many Western economies (e.g., Italy, UK, US) suffered from slow rollouts of COVID-19 interventions due to politically-driven and -managed government regulations and healthcare delivery systems, Asian governments (e.g., Singapore, South Korea, Taiwan, Vietnam, Hong Kong SAR) were quicker to centralize disease control and vaccine rollouts by bypassing bureaucratic structures of healthcare administration [25–27]. Institutional differences played a part in the speed and the adoption of the COVID-19 intervention measures [28], with the Western regimes built on the neoliberal policies of deregulation and privatization while many Eastern regimes are still relatively more centralized and regulates. Another difference lied in the West's hospital-centered public health strategies versus the East's community-centered strategies in rapid testing, treatment, and care rollouts [27,29]. Such East-West differences in the speed, scale, and channels of COVID-19 interventions, as well as the varying rates of public's adoption of prevention/treatment tools, would have affected lay people's labor force participation, production activities, and acceptance of WFH arrangements.

Singapore, a Southeast Asian city state, was among the first to be hit by the pandemic with cases reported in January 2020. In Singapore, the process of adopting remote work and fighting against COVID-19 could have differed from that of the Western countries due to its distinct labor market structure—with more than 85.0% of male and female Singaporean residents working full-time, a majority of them working in the high-income white-collar occupations (e.g., managers,

administrators, professionals, technicians), and over 80.0% of both genders working in services industry in 2020 [8]. Further information on the Singapore's labor market is provided in the S1 Table and S2 Table. Our study constitutes one of the first studies in Singapore estimating the impact of WFH during the COVID-19 induced lockdown on labor market outcomes, complementing the qualitative evidence which suggested that the share of respondents reporting productivity increases following the introduction of remote work rose from 15.1% in April 2020 to 22.5% in June 2020 [30].

Singapore serves as a useful setting for studying the labor market impacts of COVID-19 for various reasons. The nation-wide lockdown mandate, observed from 7 April 2020–1 June 2020, required workers whose occupational tasks could be carried out remotely to work from home regardless of personal preferences [31]. The mandate was strictly enforced, thereby alleviating issues of self-selection into remote work arrangements. Moreover, the country experienced a relatively smooth economic recovery and transition in- and out- of the lockdown during 2020, unlike many other economies whose experiences were marked by sporadic virus outbreaks and lockdowns. By the last quarter of 2020, the rate of decline of GDP slowed down significantly and started bouncing back to pre-pandemic activity levels in the following quarters, coinciding with a steep decline in number of daily confirmed cases to mostly single digits (**Fig 1**). Employment growth rate turned positive around the same time, making it easier to identify the disparate impacts of government restrictions on the income trajectories of remote and non-remote workers.

We organize our study as follows. Background Section provides an overview of the international literature on the implications of remote work for gender inequality before and during the pandemic, implications of the adoption of remote work on gender equality, as well as the Singaporean context (i.e., gender norms, division of household labor). Data and Methods Section presents the data, outcome variables, covariates, and descriptive statistics of our sample. Estimation Section explains the empirical strategies. Results Section presents our main findings on the impact of adoption of remote work on income, wages, and work hours as well as differences in the impact of remote work across male and female spouses, and mediating effects of household responsibilities. Discussion Section evaluates the implications and limitations of our results, and Conclusion Section considers policy implications of the findings.

## 2. Background

### 2.1. Remote work and labor market outcomes

In the pre- COVID-19 periods, remote work arrangements are often seen to generate positive outcomes for both firms and workers. A study by Bloom et al.[32] using data from a Chinese call center provides evidence that remote work options increase workers' performance and job satisfaction. This finding is supported by qualitative studies conducted in Australia, Belgium and Japan, which show that remote work can enhance worker productivity by providing flexibility, reducing commuting costs, and minimizing interruptions at the workplace [33–35].

In the COVID-19 era, we have seen a surge of studies around the time of the pandemic that highlight nuanced effects of the remote work arrangements on the labor market outcomes. A stream of literature shows that the workers' ability to adopt remote work arrangements alleviated the negative shock of the pandemic such as layoffs [36], reduced work-family conflicts [[37], increased productivity [38], and promoted job satisfaction [39]. In contrast, job losses as well as physical and mental tolls were especially severe in occupations that could not adopt remote work arrangements and had to maintain physical contacts with their customers [40,41]. In the case of Singapore, a survey finds that as of April 2021, 85% of employees and 70% of managers *believe* that workers were more or equally productive working from home as in the office [42]. In this survey, respondents are overwhelmingly in favor of maintaining flexible work arrangements, with 60–70% preferring a hybrid work. Similarly, around three-quarters of respondents in our dataset stated that they are strongly or somewhat in favor of permanent work from home arrangements (see S1 Text, S1 Table, and S2 Table).

Yet, another stream of literature points out that workers' preference toward remote work (over office work) may not be strictly driven by their expectation of better labor market outcomes. Such preferences as well as the productivity

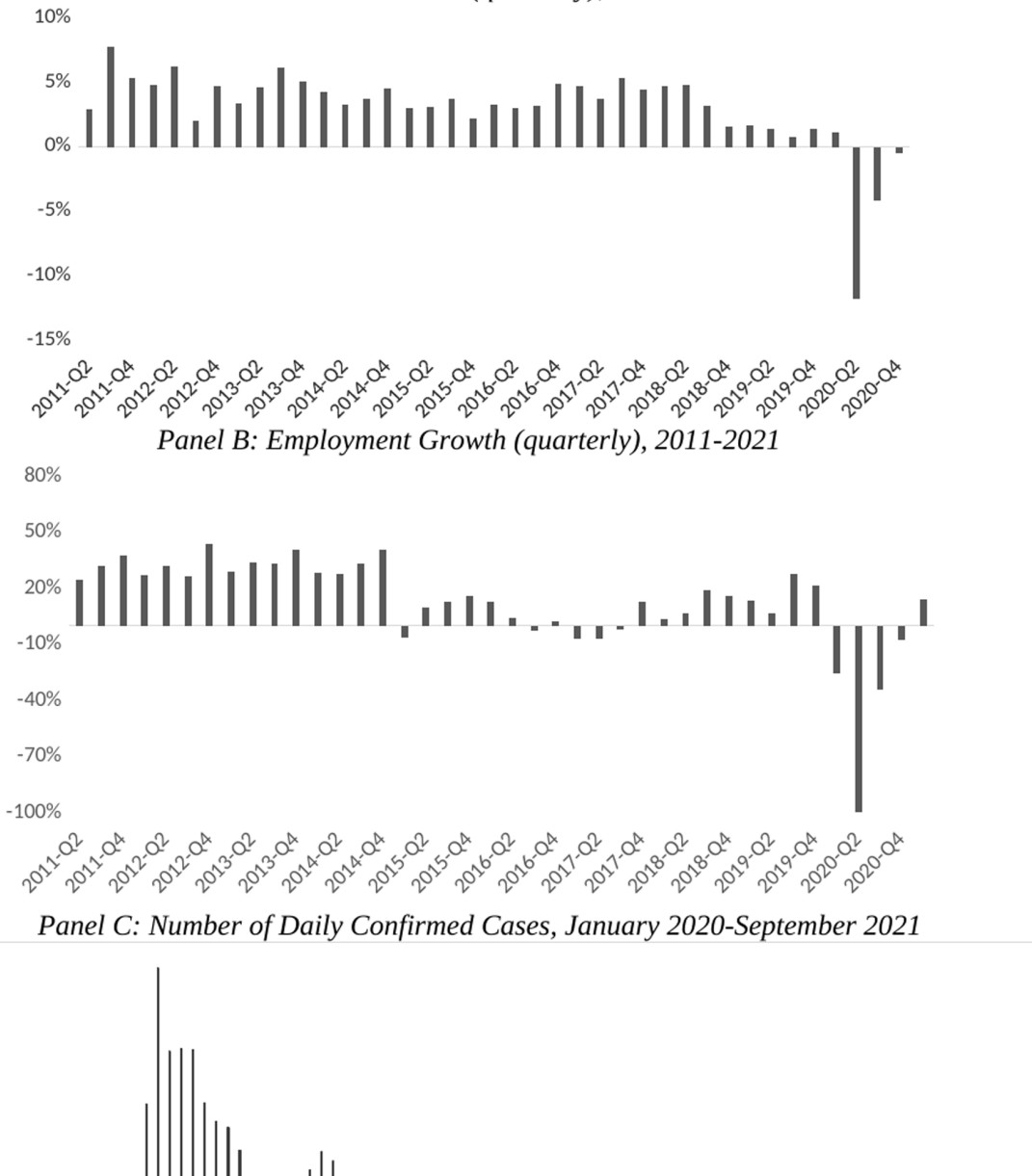

**Fig 1. GDP, Employment Growths, and COVID-19 Cases in Singapore, 2020-2021.**

differential between home-based versus office-based work may arise due to individuals' *varying abilities to sort into the modes of work*. Atkin et al. [43] shows that workers who prefer to work from home show a substantially lower productivity when working from home as opposed to in the office, using data from the data entry sector in India. Despite the deteriorating outcomes, such individuals are constrained to choose remote work over the office-work due to household responsibilities such as childcare. When the self-sorting is removed, and when workers are randomly assigned into home-based versus office-based arrangements, workers who are assigned to work remotely show lower productivity than those in the office—thereby contrasting the findings in the earlier studies [39]. Such negative impact of remote work is further substantiated by Gibbs et al. [44], who show that the remote work lowered productivity (i.e., increasing work hours while lowering outputs) among the skilled workers in a technology company in India. Heavy communication- and coordination- costs are the main culprits of the productivity declines.

Labor market outcomes can also vary depending on the type of remote work arrangements adopted. For instance, the availability of remote work infrastructures has paved way for 'hybrid work' as a popular mode of work—whereby employees divide their hours between working in the office and at home. While the hybrid work increases job-satisfaction, lower attrition, and is expected to boost productivity among the employees, such an arrangement is predicted to lower productivity by the managers [45].

## 2.2 Remote work and gender inequality

Against the backdrop of potential gains from remote work, some studies have focused on the implications for gender gaps in labor market outcomes. Generally, women and lower-income workers have been disproportionately affected by the pandemic, partly due to their higher representation in the most ravaged sectors such as the service industries, e.g., travel and hospitality [3,14], and partly due to their holding jobs with low remoteability [41,46,47]. Women also have been seen to face greater job losses even after accounting for differences in job types and remoteability [2].

One of the drivers of the gender gap lies in the underlying *disparities in division of household labor* [22,48], which are further exacerbated by the rise of remote work and blurred boundaries between work and household duties in the COVID-19 era [15]. Studies consistently find that mothers rather than fathers bore the brunt of increased domestic work demands during lockdowns even if both were working full-time, with one paper estimating that women spent as much as 40% more time on childcare [14,20–24]. In response to the sudden spike in childcare and household responsibilities due to closure of schools and childcare facilities and loss of access to nannies and other helpers, women were more likely to leave employment or cut work hours, even if their work could be done remotely [14,21,49,50]. While much of the evidence comes from the Western context, this phenomenon was also observed in Singapore: While the employment rate did not drop considerably between 2019 and 2020 due to nation-wide COVID-19 support measures such as Jobs Support Schemes (JSS) for wage support or SGUnited Jobs and Skills Package and Jobs Growth Incentives for continued hiring [51], female Singaporean employed residents experienced greater reduction in working hours or more frequent implementation of temporary no-pay leaves, compared to men in 2020 [8].

Because mothers in dual-income families tended to spend more time than fathers on childcare prior to the pandemic, children generally expected and preferred care from their mothers during lockdowns, resulting in women performing parenting tasks even when fathers or other caregivers were available [15,49]. In spite of the exacerbations in pre-existing gender gaps in the labor market outcomes evidenced in the recent literature, Alon et al.[14] argue that the rise of the flexible work arrangements could eventually improve gender equality by inducing further paternal involvement in childcare following the parents' adoption of remote work [16], or by providing women with greater schedule control as remote work options enable spatial and temporal flexibility [17,52].

The streams of literature, taken together, suggest potentially divergent labor market experiences of male and female spouses under COVID-19 following the adoption of remote work arrangements. Moreover, the literature highlights the importance of household resources and responsibilities as mediators in determining whether married men and women will benefit equally from remote work.

The gendered nature of the labor market experiences and of the effects of remote work arrangements is especially salient in the context of Singapore, where gender norms tend to be socially conservative. Although it is common for married women to hold a full-time job alongside their husbands, they are still expected to be primarily responsible for domestic labor and care work [53], resulting in high levels of physical, mental, and emotional exhaustion [54]. Moreover, the strong societal emphasis on children's academic achievement has created very stressful conditions for parents, especially for mothers [53]. Consequently, mothers spend more time with their children than fathers do on average, with one study estimating that employed and stay-at-home Singaporean mothers spend an average of 3.2 and 5.6 hours respectively with young children on weekdays, whereas fathers spend 1.75 hours regardless of work status [55].

Further adding to the problem of the unequal division of household responsibilities across gender is the declining resources for childcare for women in Singapore. The rapid demographic and economic transformation Singapore has undergone—which, in turn, increased individuals' earnings and preferences for separate living spaces [56]—reduced three-generational households from 11.3% to 8.7% between 2010 and 2017 [57]. As a result, married women with young children are increasingly unable to rely on elderly parents or in-laws for childcare assistance. In place of intergenerational exchanges of physical care and help with domestic tasks, hiring a live-in female low-wage domestic worker from a neighboring country such as Indonesia or the Philippines has become an increasingly common practice, with official statistics showing an increase of over 10% from 231,500–261,800 between 2015 and 2019 [58]. However, at the societal level, the increasing availability of foreign domestic workers has lowered the status of domestic work and social relevance of women's struggles to achieve work–life balance [59].

During the COVID-19 Lockdown from 7 April 2020–1 June 2020, the pre-existing living arrangements and division of household responsibilities have become even more fundamental drivers of the labor market participations/outcomes of married spouses—as adjustments to the arrangements (e.g., securing external childcare support) have become unlikely. During this period, schools and childcare centers were closed as the law mandated that only "essential" businesses and services such as supermarkets, delivery services, food suppliers, energy manufacturers and medical centers remained open [31]. Households were not allowed to reach out to family members from separate dwellings [60], and households who had not hired foreign domestic workers prior to the lockdown faced difficulties doing so during the height of the crisis [61]. Households were also not allowed to hire ad-hoc services to help with household tasks apart from essential services such as food delivery. After the lockdown ended in early June, up to two daily visitors from the same family (including grandparents) were permitted, but engagement of non-essential household services was still prohibited [62]. Violations of lockdown guidelines resulted in fines up to SGD $10,000 (approximately $7,500 USD) and/or up to six months' imprisonment.

### 2.3 Hypotheses

Based on the insights from the literature on the links between remote work and labor market outcomes, gendered nature of the linkages, and Singapore's contextual specificity, we propose to test the following three hypotheses in this paper:

**Hypothesis 1** Compared to individuals who worked outside, individuals who were able to shift to remote work during the lockdown period experienced monthly income gains **due to increased productivity**— facilitated potentially by added flexibility, reduced disruption from peers, and reduced (psychological) burden of commuting. The gains persisted over time.

**Hypothesis 2: Monthly Income and productivity gains** from adopting remote work were lower for female spouses than for male spouses. The gender gap in gains from remote work diminished after the lockdown ended, as childcare facilities and other services became available.

**Hypothesis 3** Individuals who spent less time on household responsibilities or who had external helpers **experienced greater monthly income and productivity gains** from working fully remotely. This is especially true for women, who tend to bear primary responsibility for unpaid household work.

Based on the insights from the WFH literature [30,33–35], we contend that the gains in monthly income induced by the adoption of WFH arrangements is driven primarily by upward changes in productivity—which are measured in terms of hourly wages in this paper. The following section details the data used to empirically evaluate the three hypotheses, key variables, and summary statistics of the final sample.

## 3. Data and methods

### 3.1. Data

A longitudinal data was drawn from our first-hand survey from Singapore between 2018 and 2020. The baseline survey (wave 1) in 2018 was in-person and in English, collected without the presence of third parties, either in nearby public spaces, or alternatively, at the respondents' homes at a time of their convenience. Participants were recruited using street intercept at central public spaces such as metro station exits, walkways of bus interchanges, spaces outside shopping malls and neighborhood town centers, stratified by the five main regions of Singapore: Central, North, Northeast, West and East. The baseline survey was intended to be a nationally representative data on married women in Singapore, collecting information on the respondents' and their spouses' sociodemographic traits, employment statuses, family composition and characteristics (e.g., household size, number of children, birth of the children, living arrangements), marriage (e.g., marital satisfaction, division of chores/childcare/ home responsibilities, marital attitudes, household attitudes), fertility (e.g., pregnancy-related questions, decision to have children, access to childcare, expectation vs reality), and health (e.g., general health, sexual habits, menstrual cycles, fertility habits, lifestyle habits). Later, we added new questions on COVID-19, labor market outcomes (monthly salary income and monthly work hours), and remote work arrangements in the subsequent online surveys disseminated in 2020—asking questions regarding respondents' current states as well as past states (i.e., with retrospective questions asking about the past waves) to assess the effect of COVID-19.

The surveys were collected three times between 2018 and 2020: (1) April-July 2018 (in-person), (2) May 2020 (online), and (3) November 2020 (online). Each recruited participant received up to SGD $20 (USD 15) for their participation in the 2018 baseline survey, SGD $25 (USD 19) for their participation in the May 2020 online surveys, and SGD $15 (USD 11) for their participation in the November 2020 online survey. The informed consent forms were signed in person during the baseline study. Participants in follow-up online surveys provided consent through online signatures. The study included no minors. This study, survey questionnaire, as well as all consent forms were pre-approved by the National University of Singapore Institutional Review Board (Ref No. LN-17–048).

The surveys disseminated 2020 included retrospective questions. For instance, the May 2020 survey asked about the respondents' conditions in May as well as in Dec. 2019 and March-April 2020—slightly before and during the imposing of the lockdown. The November 2020 survey asked about conditions in November as well as in June 2020—shortly after the lockdown was lifted. Taking advantage of the temporal variations in the questions we asked, we constructed a total of 6 waves in our estimations: April-July 2018 (wave 1), December 2019 (wave 2), March-April 2020 (wave 3), May 2020 (wave 4), June 2020 (wave 5), and November 2020 (wave 6). Wave 2, 3, and 5 were reverse-engineered. The timeline for each wave of data collection in proximity to major events during the pandemic is shown in **Table 1**.

The survey followed 660 female participants—all of whom were married, Singaporean residents, and were able to read, write, and speak in English—who reported information on themselves as well as their 660 spouses (all male). As we pulled the information on the respondents' spouses into distinct observations, our data consisted of a total of 3,960 person-wave observations for each gender in 6 waves, all of whom fell between ages 24 and 59 during the analytic period. Among the total of 7,920 person-wave observations, 2,796 were unemployed or out of the labor force (1,269 men and 1,527 women). This suggests that 2,691 male respondents (i.e., 67.95% of the total of 3,960 male person-wave observations) and 2,433 female respondents (61.44% of the total of 3,960 female person-wave observations) were employed. Compared to the 88.94% employment among the male population aged 25–59 and 76.20% employment

**Table 1. Key Dates of the COVID-19 Pandemic and Survey Data Collection.**

| | April-July 2018 | Dec 2019 | Jan 2020 | Mar-April 2020 | May 2020 | June 2020 | Nov. 2020 |
|---|---|---|---|---|---|---|---|
| Pandemic timeline | Pre-crisis | | Singapore confirms first infection case on Jan 23 | Circuit breaker begins on 7 April | Circuit breaker cont. | Circuit breaker ends on June 1; Phases I/II commence | Phase II with relaxed restric-tions |
| Data collection timeline | In-person survey | | | | Online survey | | Online survey |
| Dates of Dissemi-nation | April 17–22, 2018 | | | | May 15–20, 2020 | | Nov 20–25, 2020 |
| Survey waves | Wave 1 (actual) | Wave 2 (reverse-engineered) | | Wave 3 (reverse-engineered) | Wave 4 (actual) | Wave 5 (reverse-engineered) | Wave 6 (actual) |

Notes:

1.The baseline survey in 2018 (wave 1) was conducted face-to-face in English without the presence of third parties, either in nearby public spaces, or alternatively, at the respondents' homes at a time of their convenience. The baseline survey and the subsequent waves collected detailed information on the respondents (all female)' and their spouses (all male).

2.Due to the unexpected emergence of the pandemic, survey waves conducted in 2020 included both current and retrospective questions regarding respondents' labor market outcomes and remote work arrangements immediately before- and shortly after- the Lockdown (Mar-April 2020). Based on all information collected across three points in time, a total of 6 waves were constructed in our data for respondents and their spouses.

3.This study, survey questionnaire, as well as all consent forms were pre-approved by the National University of Singapore Institutional Review Board (Ref #: LN-17–048). Section 3.1 **Data** contains further information.

among the females reported in the Singapore's official statistics in *2020 Labour Force in Singapore* [63], our data under-sampled employed persons.

From the 7,920 person-wave observations, we dropped 68 observations who were separated or divorced, leaving 655 men and 655 women in waves 1–5, and 651 each in wave 6. The continuous marital status was important in that, if a respondent was divorced in a survey wave, this person would have no response to provide for the husband. Because we explored the role of household work division across spouses as mediators in this study, we required the spouse information. Next, we dropped 3,184 observations (all observations for 531 individuals across waves) for not working in waves 1–2 in 2018–2019, prior to COVID-19 and Lockdown, as changes across time in their monthly income or hourly wages could be measured. We further dropped from our final sample 360 person-wave observations for individuals who worked in waves 1–2 but *stopped working* in waves 3–6 in 2020 and individuals who stayed *unemployed/out of the labor force* for all 6 waves. This was to distinguish the effect of remote work adoption from that of job displacement/ acquisition across waves. Including individuals who did not work in 2018–2019 and started working in 2020 would have introduced an upward bias to the effect of remote work arrangements. If such individuals engaged in WFH jobs in 2020, their income and wages would have experienced a steep jump from zero. This would not necessarily be a labor mar-ket outcome associated with the adoption of WFH arrangements. Rather, this would be the outcome associated with transitioning from the non-working state to the working state. Similarly, including in the sample individuals who worked in waves 1–2 but stopped working in 2020 would have generated a downward bias. Lastly, individuals without jobs in 2020 would not be able to be categorized based on the extent of WFH they have adopted (i.e., our main regressor). In short, we intentionally kept only the employed individuals to measure the effect of adopting remote work arrangements *conditional on individuals' continued employment throughout the analytic period.* Our sample accommodated switching between jobs.

In the end, our final sample consisted of 4,308 person-wave observations in an unbalanced panel with 384 men (in waves 1–5, with a subsequent exclusion of 3 respondents in wave 6) and 335 women (in waves 1–5, with a subsequent exclusion of 3 respondents in wave 6)—all of whom were married and employed during the entire 6 waves.

## 3.2. Key Variables

The main outcome variables we examined are monthly salary income, monthly hours worked, and hourly wages. Of note, the monthly income was measured in brackets: no income, less than SGD $1,000, $1,000–1,999, $2,000–2,999, $3,000–3,999, …, $9,000–9,999, and $10,000 or more. To simplify the interpretations of our estimations, we used mid-points of each income bracket except for the last bracket, which is replaced with SGD $15,000. Using the information on the number of hours worked per month, we calculated the hourly wages, i.e., a proxy for productivity.

Next, we collected information on the extent of remote work individuals performed: indicators for whether working fully from home, mostly from home, half from home and half outside the home, mostly outside the home or fully outside the home. While the lockdown required that the adoption of remote work for individuals whose job could be done fully from home, the rest of the job-holders experienced varying rates of WFH operations depending on the occupational tasks. Also, even if an individual could be fully remote for one job, this person could still work partially outside for the second job if he or she held multiple jobs—about 2.8% of employed persons in Singapore in 2020 [63]. Even though our survey did not ask about the number of jobs held in any wave, the practice of moonlighting was and continues to be on a steady uptrend since 2012 and especially during the pandemic as the Singapore's National Wage Council and the Ministry of Manpower encouraged the practice in their wage and training guidelines and an advisory report on second-job arrangements, respectively [64,65]. As a main explanatory variable, we created a binary indicator of whether individuals were working fully from home. For the binary variable, we intentionally used a rigid definition of remote work to distinguish the labor market outcomes of individuals who fully transitioned into working from home from that of the rest (no WFH or partial WFH)—who would have been subject to more work-related disruptions or health-risks following the COVID-19 outbreak. We tested the implications of using a less rigid definition of remote work in the independent variable in S8 Table in S2 Text.

For time indicators, we grouped the 6 waves into three meaningful time periods: pre-COVID-19 outbreak (April-July 2018 and December 2019); the lockdown period and immediately surrounding months (March-June 2020); and the post-lockdown recovery phase (November 2020). The different periods are intended to evaluate the immediate and longer-term impacts of the adoption of remote work—thereby operationalizing *Hypothesis 1.*

Because our survey asked questions regarding the female respondents themselves as well as their spouses (all male), we used the responses on the male spouses as distinct observations (see Section 3.1 for details). Such extensions of our sample allowed us to evaluate differences in gender and across spouses—operationalizing *Hypothesis 2.*

For covariates, we collected information on time-varying individual and household characteristics, namely number of children and a binary indicator for presence of an infant aged two or below, and dichotomous indicators for occupation. Of note, the survey provided 29 occupational categories, embedded under 4 larger groupings: Professionals, Associate Professionals and Technicians, Clerical, and Service/Sales (See S3 Table for detailed categorization).

We considered the following sets of mediators to explore the heterogeneity of the impact of COVID-19 on income, wage, and work hours across remote and non-remote workers: a) time spent on childcare and household chores (above or below median levels of time use for each category), b) presence of a potential household helper, and c) spouse's remote work status. Occupation types reflect differences in workers' cognitive and skill levels, which often drive productivity differentials [46,66]. The mediators representing household resources and responsibilities were needed to operationalize *Hypothesis 3,* which could have affected workers' ability to benefit from remote work arrangements. Time use in childcare and household chores were measured based on responses to two questions: "How many minutes per day do you/your husband spend on the following? A) household chores like cooking and cleaning (exclude childcare), B) caring for children (can include feeding and bathing or reading to them)." These variables were recorded as number of minutes per hour spent in current time t, after confirming non-mechanical links between childcare/chore and WFH, i.e., an increase in time spent in childcare/chores did not automatically lead to a decrease in the time spent on WFH, and vice versa. Detailed discussions are provided in S9 Table and S10 Table in S3 Text. Presence of an external helper was measured as a dichotomous indicator for whether at least one of the following reside in the household: a parent, parent-in-law and hired

domestic helper, who could provide informal childcare in place of the unexpected loss of access to formal facilities. Similarly, spouse's job remote work status indicates whether he/she might be physically available to take on household tasks. All moderating variables are coded as binary (1 if available, 0 if otherwise).

### 3.3. Summary statistics

Our final sample consisted of 4,308 person-wave observations—2,301 males (tracking 384 men) and 2,007 women (tracking 335 women)—between ages 24 and 59. Summary statistics for the male and the female subsamples, further divided into pre-lockdown, lockdown, and post-lockdown phases, are displayed in **Table 2**. Around 90% of the sample were ethnic Chinese. The average ages of male and female observations were around 35 and 32 respectively. In the pre-lockdown period, about 25–27% of male and female spouses had no child; 41–43%, one child; 25–26%, two children; and 5–6%, three of more children. While the share of respondents with no children or one child decreased in the lockdown and post-lockdown periods, that of those with two or more children increased. The share of respondents having an infant aged two years or younger jumped from 1% in pre-lockdown to about 40% during lockdown/post-lockdown—which could have been partially driven by policies introduced in 2019 and 2020 to encourage childbirth, including the enhanced CPF Housing Grant for first-time homebuyers encouraging family formation [67], the Intra-Uterine Insemination co-funding scheme to financially support couples facing fertility challenges [68], increased preschool subsidies [69], and a one-time Baby Support Grant [70].

Although women were more likely to have a college degree (72% of women vs. 64% of men), monthly salary income was higher among men—with men on average earning around SGD $5,400 monthly income during the pre-lockdown period (jumping to SGD $5800 in 2020) and women on average earning about SGD $4,200 (jumping to SGD $4400 in 2020). While some of the gender income- gaps could be explained by differences in the monthly hours worked, with men working an average of 187 hours per month and women working 170 hours per month in pre-lockdown, similar gender gaps were seen in their hourly wages (SGD $31 hourly wages for men in pre-lockdown, jumping to SGD $40 in 2020 vs. SGD $26 hourly wages for women, jumping to SGD $33–38 in 2020).

In the table, we showed average hourly wages, monthly income, and monthly work hours by WFH statuses. In all three time periods (pre-lockdown, lockdown, and post-lockdown), WFH workers earned more than non-WFH workers both in terms of monthly salary income and hourly wage. Across gender, we confirmed that the gender gaps in monthly income and hourly wages persisted across WFH and non-WFH statuses. Moreover, for both genders, while the average hourly wages increased during the lock-down for both WFH and non-WFH workers, the average monthly income increased under the lockdown for only the WFH workers and *not* for non-WFH workers. The increased monthly income of WFH individuals was not driven by increased monthly work hours.

Across the three periods, more women worked fully remotely than men (50% vs. 35%)—with both shares dropping significantly in the post-lockdown periods. The higher proportion of female WFH workers may reflect the difference in occupational distributions, and are in line with their preferences (S4 Table). For instance, while similar shares of men and women were in professional (about 80%) jobs, more women were in associate professional jobs (12% among women and 9% among men) and in clerical jobs (8% among women and 2% among men) which could be easily adopt full remote work arrangements, as opposed to service jobs. More men found themselves in services jobs (12–19% as opposed to 6–13% women across the periods), which required more face-to-face contact. Across genders, the percentage of professionals and associate professionals decreased during the lockdown. Yet, the share of service workers increased during lockdown, which may be a result of the former professionals and associate professionals entering the services industry during the pandemic as this industry was well-protected (e.g., wages were 50–70% subsidized) by the government under the Jobs Support Scheme [51, 71].

Both genders had greater shares of full WFH workers in the pre-lockdown phase (38% male and 56% female), with the shares dropping slightly (35% and 52%, respectively) during the lockdown. As we studied the effect of WFH arrangements

**Table 2. Descriptive Statistics.**

| | Male | | | Female | | |
|---|---|---|---|---|---|---|
| | Pre-Lockdown | Lockdown | Post-Lockdown | Pre-Lockdown | Lockdown | Post-Lockdown |
| *Outcome variables* | | | | | | |
| Hourly wages (SGD) | | | | | | |
| for everyone | 30.72 † | 41.08 | 39.54 † | 25.88 | 38.14 | 33.04 |
| | (21.66) | (54.12) | (44.70) | (16.63) | (69.64) | (34.07) |
| for full WFH | 34.95 † | 50.17 | 61.37 † | 27.83 | 44.83 | 41.57 |
| | (18.10) | (50.87) | (59.86) | (19.89) | (88.03) | (52.50) |
| for partial or non-WFH | 28.19 † | 36.28 † | 37.20 † | 23.38 | 31.01 | 31.08 |
| | (23.20) | (55.19) | (42.20) | (10.68) | (40.81) | (27.98) |
| Monthly income (SGD) | | | | | | |
| for everyone | 5471.35 † | 5409.72 † | 5761.15 † | 4197.76 | 4195.52 | 4382.53 |
| | (3043.12) | (3224.53) | (3194.47) | (1991.03) | (2174.03) | (2256.19) |
| for full WFH | 6275.86 † | 6604.27 † | 7000.00 † | 4410.90 | 4510.60 | 4645.16 |
| | (3248.60) | (3591.88) | (3704.35) | (2165.03) | (2443.45) | (2761.54) |
| for partial or non-WFH | 4983.26 † | 4779.18 † | 5627.91 † | 3925.17 | 3859.05 | 4322.22 |
| | (2804.37) | (2817.60) | (3111.57) | (1708.60) | (1785.45) | (2124.79) |
| Monthly Hours Worked | | | | | | |
| for everyone | 187.45 † | 174.70 † | 183.21 † | 170.36 | 159.74 | 165.23 |
| | (45.91) | (77.82) | (73.28) | (46.24) | (79.56) | (80.50) |
| for full WFH | 181.92 † | 164.36 | 161.30 | 170.43 | 155.11 | 139.87 |
| | (29.01) | (66.22) | (75.37) | (57.27) | (91.01) | (79.97) |
| for partial or non-WFH | 190.80 † | 180.16 † | 185.57 † | 170.28 | 164.68 | 171.05 |
| | (53.38) | (82.82) | (72.77) | (26.16) | (64.88) | (79.64) |
| *Independent variable* | | | | | | |
| Working Fully from Home (%) | 37.76 † | 34.55 † | 9.71 † | 56.12 | 51.64 | 18.67 |
| *Moderating variables* | | | | | | |
| Occupation (%) | | | | | | |
| Professional | 86.20 | 79.08 | 79.00 | 83.58 | 78.81 | 78.61 |
| Associate professional | 9.11 | 8.85 † | 8.92 | 8.96 | 11.94 | 12.05 |
| Clerical | 1.82 † | 1.56 † | 1.57 † | 8.36 | 8.06 | 8.13 |
| Service | 11.98 † | 19.27 † | 19.16 † | 5.67 | 12.84 | 12.95 |
| Other or unknown | 0.01 † | 0.09 | 0.26 | 2.39 | 0.30 | 0.30 |
| Time use on childcare (min/hr) | | | | | | |
| for everyone | 3.40 † | 5.62 † | 4.56 † | 7.24 | 10.33 | 7.20 |
| | (4.83) | (8.58) | (6.57) | (11.81) | (11.61) | (8.85) |
| for full WFH | 3.70 † | 6.10 † | 7.21 | 7.52 | 11.08 | 9.86 |
| | (5.87) | (8.28) | (7.69) | (10.30) | (11.70) | (11.71) |
| for partial or non-WFH | 3.21 † | 5.37 † | 4.28 † | 6.88 | 9.52 | 6.59 |
| | (4.07) | (8.72) | (6.39) | (13.50) | (11.47) | (7.95) |
| Time use on chores (min/hr) | | | | | | |
| for everyone | 1.57 † | 2.48 † | 2.02 † | 2.34 | 4.47 | 3.13 |
| | (2.32) | (3.83) | (3.06) | (2.77) | (6.11) | (3.36) |
| for full WFH | 1.38 † | 2.44 † | 1.70 † | 2.16 | 4.32 | 2.62 |
| | (1.52) | (3.08) | (1.64) | (2.43) | (5.51) | (2.09) |
| for partial or non-WFH | 1.68 † | 2.51 † | 2.05 † | 2.56 | 4.64 | 3.24 |
| | (2.68) | (4.18) | (3.17) | (3.15) | (6.70) | (3.58) |

*(Continued)*

**Table 2.** (Continued)

| | Male | | | Female | | |
|---|---|---|---|---|---|---|
| | **Pre-Lockdown** | **Lockdown** | **Post-Lockdown** | **Pre-Lockdown** | **Lockdown** | **Post-Lockdown** |
| Spouse's time use on childcare (min/hr) | 9.18 † | 11.51 † | 8.86 † | 3.32 | 5.81 | 4.33 |
| | (14.39) | (12.42) | (10.47) | (4.62) | (9.08) | (6.67) |
| Spouse's time use on chores (min/hr) | 2.82 † | 4.92 † | 3.44 † | 1.57 | 2.77 | 2.07 |
| | (4.19) | (6.25) | (3.43) | (1.87) | (4.46) | (3.26) |
| Presence of domestic help (%) | 10.94 | 26.65 | 27.30 | 11.04 | 29.25 | 29.22 |
| Co-residence with parents (%) | 12.50 † | 12.07 † | 11.29 | 67.16 | 23.58 | 15.96 |
| Co-residence with parents-in-laws (%) | 69.27 † | 22.22 † | 14.96 | 14.03 | 13.53 | 12.65 |
| Spouse working remotely (%) | 47.66 † | 44.27 † | 17.59 † | 38.96 | 34.13 | 8.13 |
| *Demographic characteristics (Covariates)* | | | | | | |
| Age | 34.03 † | 35.53 † | 35.53 † | 31.25 | 32.75 | 32.75 |
| | (3.73) | (3.70) | (3.71) | (2.53) | (2.48) | (2.48) |
| Race (%) | | | | | | |
| Chinese | 91.41 | 91.41 | 91.86 | 93.13 | 93.13 | 93.37 |
| Indian | 1.30 | 1.30 † | 1.31 | 2.39 | 2.39 | 2.11 |
| Malay | 4.69 | 4.69 | 4.46 | 4.48 | 4.48 | 4.52 |
| College educated (%) | 64.06 † | 64.06 † | 64.04 † | 71.64 | 71.64 | 71.39 |
| Number of children (%) | | | | | | |
| No child | 25.26 | 13.02 | 12.07 | 27.16 | 14.63 | 13.55 |
| One child | 43.23 | 33.59 | 29.92 | 41.49 | 33.63 | 30.42 |
| Two children | 26.04 | 43.40 | 46.72 | 25.67 | 42.19 | 45.48 |
| Three or more children | 5.47 | 9.98 | 11.29 | 5.67 | 9.55 | 10.54 |
| Presence of infant below age 2 (%) | 1.17 | 38.63 | 45.14 | 1.19 | 37.21 | 43.07 |
| Total person-wave observations | 768 | 1152 | 381 | 670 | 1005 | 332 |
| full WFH subsample | 290 | 398 | 37 | 376 | 519 | 62 |
| partial or no WFH subsample | 478 | 754 | 344 | 294 | 486 | 270 |

*Notes: WFH stands for work-from-home. The outcome variables and time use variables are disaggregated by remote work statuses. Of note, 'WFH individuals' refer to those working fully remotely from home, while 'non-WFH individuals' refer to those working at least partially outside. Numbers in parentheses are standard deviations.*

†*+ denotes that the male-female differences in the average or percent--for each analytic period--are statistically significant (p<0.05).*

on the labor market outcomes *conditional* on that individuals stayed employed throughout the analytic period, a possible reason for the switch from full WFH to non-WFH (i.e., working at least partially outside) states could be due to job switching *within the same broad occupational category*. In our final sample, among the total of 267 unique respondents—1,602 person-wave observations in 6 survey waves—transitioned from WFH to non-WFH states in 2020, none of them switched occupational categories. Out of 267 unique individuals, 141 went from being fully WFH in the pre-pandemic period to working mostly from home. 73 out of 267 individuals transitioned from being fully WFH to working half from home and half outside; 24 out of 267 respondents moved from full WFH to working mostly outside; lastly, 29 out of 267 respondents transitioned from full WFH to working fully outside. Once again, all these individuals stayed within the same occupational categories as in the pre-pandemic period. The occupational distributions of individuals who switched (1) from fully WFH to mostly WFH, (2) from full WFH to half-home-half-outside, (3) from full WFH to mostly outside, or (4) from fully WFH to fully outside were similar to one another. A plurality of each group consisted of individuals in white-collar occupations (e.g., in business and administration professions, ICT professions, clerical, and legal, social, and cultural professions), even though the groups that transitioned to more non-WFH states included some individuals in sales, customer services,

healthcare, hospitality, and production. It could have been that there were demand-driven reassignment of manpower within companies or reconfiguration of job tasks to account for COVID-19 related labor market hardships (e.g., manpower shortages in certain industries and certain occupations) that changed the extent to which respondents could engage in full WFH arrangements. Alternatively, it could have been that respondents started working multiple jobs to financially support themselves in 2020—a practice encouraged by the government of Singapore [64] Unfortunately, we were not able to check this because we did not directly ask in the survey the number of jobs held at each point in time.

We also show the distribution of workers by more disaggregated WFH arrangements in S11 Table. It can be seen that, beside the full WFH category, a similar share of men and women belonged in all other remote work arrangement categories—except in the 'working fully outside' category in which more men belonged than women (24% vs. 11%).

Male spouses spent much less time than female spouses on childcare (3.4–5.6 minutes per hour across the three periods for men; 7.2–10.3 minutes per hour for women) and household chores (1.6–2.5 minutes per hour across the three periods for men; 2.3–4.5 minutes per hour for women). With the lockdown, the amount of time spent on childcare and chores increased for both genders. Interestingly, the gender gap in the time devoted to childcare reduced between the lockdown and the post-lockdown periods, while that in the time spent in chores increased. Around 10–11% of the respondents report living with a foreign domestic worker, and this share increased to 27–30% in 2020 for both genders. Generally, more respondents lived with female spouses' parents (69%) than with male spouses' parents (12%) in the pre-lockdown period—reflecting the societal perception of domestic responsibilities as women's work and a tendency for married women to prefer additional help with childcare from their own parents. While the arrangements for those living with male spouses' parents did not change during the lockdown/post-lockdown periods, the share of respondents living with female spouses' parents dropped from 69% to 22% during lockdown, and to 15% during post-lockdown.

In comparing our sample to the published nationally representative statistics (e.g., from Ministry of Manpower) for the same age groups (24–59) (see S5 Table), we found that the distributions of birth parities were highly comparable. This suggested that our sample was fairly representative of the population of married women in this age range in Singapore. Our sample statistics for income and occupation distributions were not representative, but this was unsurprising as national statistics for the same age groups (24–59) were not specific to marital status. The partial representativeness of our sample is further discussed in Section 6.

## 4. Estimation

We estimated the effect of the WFH arrangements during the government-mandated lockdown on the labor market outcomes using a difference-in-differences (DID) estimation model. One of the key assumptions of the DID framework was parallel pre-trends in the outcome of interest between remote and non-remote workers. We went beyond assessing the graphical evidence and conducted a formal regression-based test to evaluate parallel pre-lockdown trends (i.e., checking whether the divergences in outcomes appeared only after the implementation of the lockdown in March-June 2020 and onward) [72]. After establishing the parallel pre-trends, we ran the following DID equation for individual $i$ holding a job $s$ at wave $t$ as our main estimations testing *Hypothesis 1*,

$$
\begin{aligned}
Y_{its} = \beta_0 &+ \beta_1 \left(Remote_{its}\right) + \beta_2 \left(Lockdown_t\right) + \beta_3 \left(Remote_{its}\right)\left(Lockdown_t\right) \\
&+ \beta_4 \left(PostLockdown_t\right) + \beta_5 \left(Remote_{its}\right)\left(PostLockdown_t\right) \\
&+ \boldsymbol{X}_{it}\beta_6 + \lambda_i + \delta_t + \gamma_s + \left(\delta_t * \gamma_s\right) + \in_{its},
\end{aligned}
\tag{1}
$$

where $Y_{its}$ referred to monthly salary income, monthly hours worked, and hourly wages, and $Remote_{its}$ was the binary indicator on WFH arrangements (1 working fully remotely, 0 working partially or fully outside). As we included in our sample individual who switched jobs across the analytic period, the within-individual variations in the adoption of remote work across waves (i.e., changes in the extent of remote work an individual faced over time) could come from any task changes

in individuals' jobs or their switching into different jobs. $Lockdown_t$ was a binary temporal indicator, taking on the value 1 if in March-June 2020 (pooled), 0 if otherwise, and $PostLockdown_t$ took on the value 1 if in November 2020, 0 if otherwise—with the reference period for both being the pre-pandemic period (April-July 2018 and December 2019, or waves 1 and 2, pooled together). $X_{it}$ was a vector of time-variant covariates, consisting of sociodemographic characteristics (age groups (24–31, 32–35, 36–59, terciled to have comparable numbers of observations in each group), number of children, and presence of an infant). We included individual fixed effects ($\lambda_i$) to account for any unobserved preferences of respondents. Further added were occupation ($\gamma_s$) and time fixed effects ($\delta_t$), thereby accounting for occupation-specificities and unobserved time-specific events. As our survey did not include questions on industry, we did not account for industry fixed-effects. As a direct consequence of the spread of COVID-19, certain occupations were more in demand than others (e.g., IT services or food delivery services). Such demand-side effects were likely to correlate with both the feasibility of WFH arrangements and the changes in labor market outcomes. In addition to measuring the extent of remote work adoption ($Remote_{its}$) at the individual level, we accounted for any structural/systematic changes a job underwent between 2018–2020 by controlling for the occupation-specific time trends ($\delta_t * \gamma_s$). $\in_{its}$ was an idiosyncratic error term. Standard errors were clustered at the household level. To explore any differences in the effect of remote work adoption across spouses, or to test *Hypothesis 2,* we re-ran all DID estimations by gender.

As a sensitivity check, we estimated the DID while replacing the binary $Remote_{its}$ with a categorical variable that differentiates individuals' work arrangements in five states (working fully outside, mostly outside, half-remote-half-outside, mostly from home, fully from home). In addition to shedding lights on the outcomes of respondents at all levels of WFH, this exercise allowed us to test whether the labor market outcomes of full WFH workers were consistently experienced by partial-WFH individuals.

Lastly, to assess the mediating roles of the household responsibilities, we included a three-way interactions among the extent of WFH ($Remote_{its}$), temporal indicators ($Lockdown_t$ and $PostLockdown_t$), and mediators (time spent in childcare and chores, availability of help, and availability of remotely-working spouses). Here, we tested whether and how the labor market outcomes of full WFH workers changed with the amount of household work borne by the spouses—thereby testing *Hypothesis 3*. Covariates and fixed effects included in this model were the same as listed above, in equation (1).

## 5. Results

### 5.1. Parallel pre-lockdown trends evaluation

Fig 2 shows the trends in monthly salary income, monthly hours worked, and hourly wages across six waves, for the purpose of graphical evaluations of the parallel pre-Lockdown trends of the (fully) WFH and non-WFH workers. Overall, the WFH and non-WFH groups followed parallel trends from April-July 2018 to December 2019 (waves 1–2, prior to COVID-19 induced lockdown) in terms of their hourly wages (panel A), monthly income (panel B), and monthly hours worked (panel C). The parallel trends assumption was satisficed for both males and females. In the pre-lockdown periods, WFH workers earned consistently higher hourly wages and higher monthly income than non-WFH workers among males and separately for females. While male WFH individuals worked fewer hours than male non-WFH individuals in the pre-lockdown period, the female WFH and non-WFH groups faced similar work hours.

Since the enactment of the lockdown (March-June 2020 and onwards), we saw a noticeable breakdown of the parallel trends in panels A-C. In panel A, the hourly wages of WFH and non-WFH workers spiked during the lockdown period—in similar rates for males and in diverging rates for females (i.e., WFH females' hourly wages increased at a faster rate than non-WFH females'). In the post-lockdown period, while the male WFH workers' hourly wages continued to increase, that of male non-WFH workers dropped. In contrast, both female WFH and female non-WFH workers experienced sharp drops in hourly wages in the same period.

In panel B, we saw that the monthly income of male WFH workers experienced a sudden increase at the start of the Lockdown (March-June 2020)—with a soft dip in May—that continued on into June and November 2020. In contrast, male

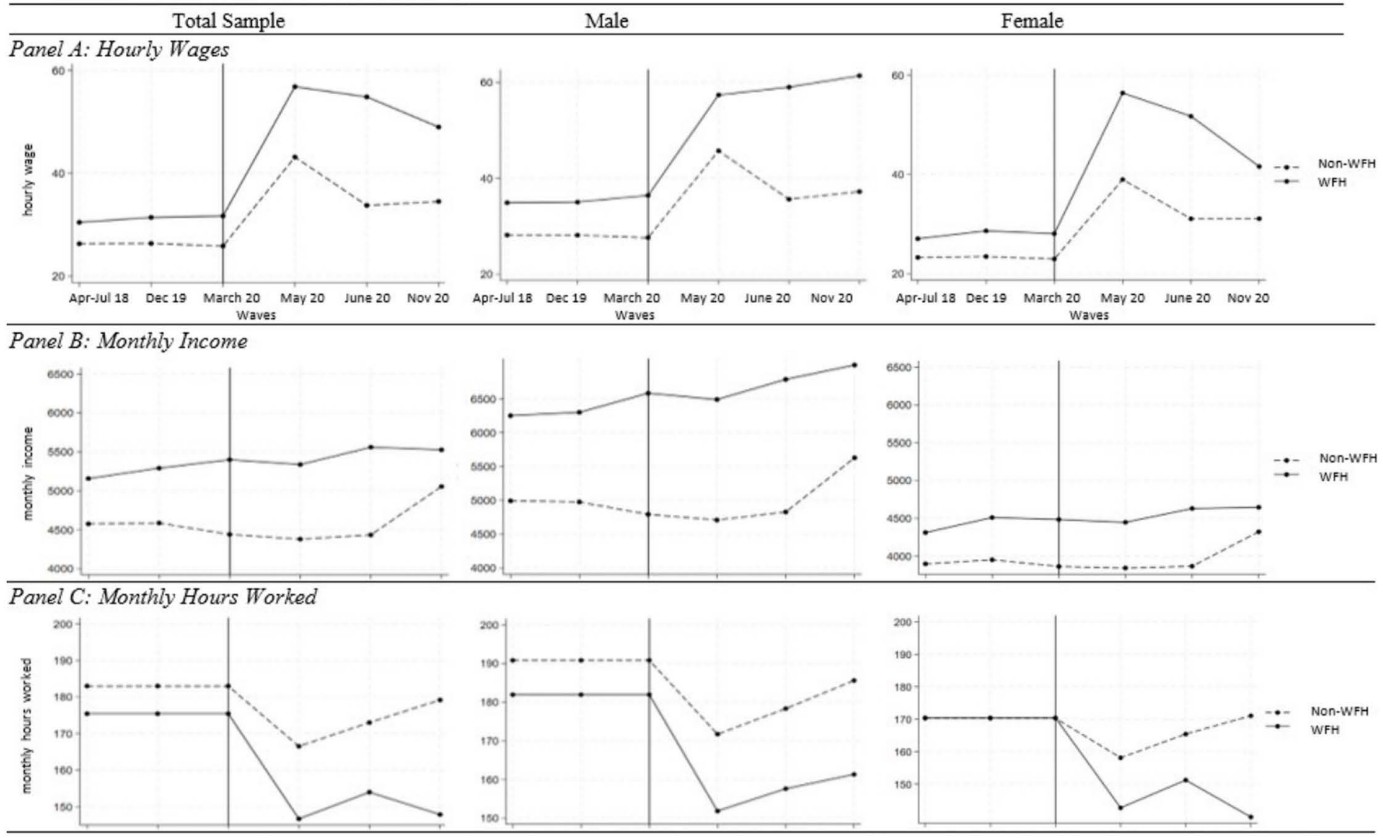

**Fig 2. Graphical Assessment of Pre-Trends.**

non-WFH workers suffered a period of stagnant or somewhat decreasing monthly income between March and June 2020 (during the lockdown period). The monthly income of female WFH workers did not increase at the start of the Lockdown (March), and started rising slightly in June 2020. But the increase was not sustained in November 2020. In short, income gains seemed to benefit male WFH workers more than their female counterparts. During the lockdown, the income trajectories of non-WFH males and non-WFH females were similar—even though the non-WFH male respondents' income underwent a sharper increase in November 2020 than that of the non-WFH women.

In panel C, monthly work hours dropped significantly for everyone regardless of their WFH statuses and gender. Yet, male non-WFH workers' monthly hours took a sharper drop than that of male WFH workers during the lockdown even though the two groups underwent a similar recovery in the post-lockdown period. female non-WFH workers' monthly hours decreased more than that of female WFH workers during the lockdown. While female WFH workers monthly hours went on a similar recovery as that of male WFH workers in November 2020, female non-WFH workers saw another sharp decrease in monthly hours in November 2020.

Next, as the graphical evaluation did not consider covariates, fixed-effects, and occupation-specific time-trends, we supplemented the graphical evidence with formal regression-based pre-trends analytics on the trends in hourly wages (**Table 3**). Analogous assessments on the trends of monthly income and monthly hours worked are available in S7 Table. The interaction terms between remote work adoption and temporal indicators evidenced the existence of parallel trends between WFH and non-WFH groups in terms of hourly wages (**Table 3**) as well as monthly income and monthly hours (S7

**Table 3. Assessment of Hourly Wage Pre-Trends.**

| | (1) | (2) | (2) |
|---|---|---|---|
| | **All** | **Male** | **Female** |
| **TREATED x TIME PERIODS** | | | |
| Working Remotely x Apr-Jul 2018 | -0.90 | -0.16 | -1.41 |
| | (0.52) | (0.85) | (0.67) |
| Working Remotely x March 2020 | 0.79 | 1.89 | -0.07 |
| | (0.68) | (0.99) | (0.80) |
| Working Remotely x May 2020 | 8.62 | 4.77 | 12.27 |
| | (6.73) | (6.84) | (10.65) |
| Working Remotely x June 2020 | 18.35*** | 18.30*** | 18.65*** |
| | (4.84) | (6.67) | (5.94) |
| Working Remotely x Nov 2020 | 15.48*** | 20.70** | 13.09** |
| | (4.82) | (8.65) | (5.54) |
| **TIME PERIODS** | | | |
| Apr-Jul 2018 | -0.05 | 0.02 | -0.18 |
| | (0.33) | (0.43) | (0.47) |
| March 2020 | -0.51 | -0.52 | -0.50 |
| | (0.39) | (0.57) | (0.42) |
| May 2020 | 16.78*** | 17.58*** | 15.48*** |
| | (3.75) | (5.04) | (5.01) |
| June 2020 | 5.79*** | 5.94** | 5.46** |
| | (1.90) | (2.58) | (2.31) |
| Nov 2020 | 4.89*** | 5.56** | 3.84* |
| | (1.85) | (2.44) | (2.25) |
| **TREATED** | | | |
| Working Remotely (1 yes 0 no) | -1.91 | -3.71 | -0.70 |
| | (3.54) | (5.13) | (4.63) |
| Individual FE | Yes | Yes | Yes |
| N | 4308 | 2301 | 2007 |

Notes: Remote is a dichotomous variable for whether the respondent worked fully from home in May 2020 during the COVID-19 lockdown. The reference time period is Dec 2019, prior to the COVID-19 pandemic. Standard errors, shown in the parentheses, are clustered at the household level. See S7 Table for the pre-trends analytics on monthly income and monthly hours worked.

Table) in the pre-lockdown period. Specifically, the two groups were not significantly different from each other during April-July 2018—relative to December 2019. All significant changes took place during the lockdown and post-lockdown periods.

While both WFH and non-WFH groups earned higher hourly wages in 2020 (**Table 2**), WFH workers' hourly wages were substantially higher—diverging from that of non-WFH workers with a lag, starting in June 2020 and onward (Table 3). The WFH workers' monthly income—for males only—was also significantly higher than that of non-WFH workers, starting in March 2020 (S7 Table, columns 1–3). Between male and female WFH workers, only males experienced significantly higher monthly income in 2020 (S7 Table, columns 1–3). In contrast, female WFH workers did not exhibit this pattern. Notably, female WFH workers also recorded significantly fewer monthly work hours than their male counterparts in 2020 (S7 Table, columns 4–6). Time fixed effects in columns 1–3 of S7 Table (for monthly income) were in line with macroeconomic data showing a large dip in economic activities during the lockdown before recovering in November 2020 (See **Fig 1**).

Discrepancies emerged when comparing results across **Table 2**, **Table 3**, and S7 Table. Specifically, while summary statistics in **Table 2** suggest that WFH workers had higher hourly wages and monthly income than non-WFH workers in

the pre-lockdown period, the regression-based results in **Table 3** and S7 Table show the opposite once individual fixed effects are included. When individuals' time-invariant characteristics are controlled in the regression-based pre-trends assessments, WFH workers had lower hourly wages and monthly income in the pre-lockdown period (Tables 3 and S7). Thus, the higher wages and income observed in Table 2 likely reflect selection effects: Individuals who opted for WFH prior to the pandemic may have been more productive or employed in higher-earning occupations. Before the mandated lockdown, WFH was a voluntary choice. Therefore, selection into WFH could have been based on individuals' innate abilities or preferences. We further investigate the links between the COVID-19 induced lockdown and WFH on the labor market outcomes in Section 5.2.

## 5.2. Effects of remote work on labor market outcomes

**Table 4** presents the DID estimates on the effect of remote work arrangements on hourly wage, monthly income, and monthly hours worked before, during, and after the government-mandated lockdown. The regressions included time-varying covariates, individual fixed effects, occupation and time fixed effects, and occupation-specific time trends, though these controls are not shown in the table. By including these fixed effects, the DID model estimates the average treatment effect on the treated (ATT), *net of* time-invariant individual traits (hereafter called, "fixed individual traits"), wave-specific

**Table 4. Effects of Remote Work Arrangements on Hourly Wage, Monthly Income, and Monthly Work Hours.**

| | (1) | (2) | (3) | (4) | (5) | (6) | (7) | (8) | (9) |
|---|---|---|---|---|---|---|---|---|---|
| | Y = Hourly Wage | | | Y = Monthly Income | | | Y = Monthly Hours Worked | | |
| | All | Male | Female | All | Male | Female | All | Male | Female |
| (reference = Pre-Lockdown) | | | | | | | | | |
| **Working Remotely x Lockdown** | 8.43*** | 7.24* | 8.73* | 272.34** | 387.43** | 65.97 † | -9.54** | -6.99 | -12.98** |
| | (3.14) | (3.71) | (5.18) | (106.74) | (178.28) | (120.13) | (3.72) | (5.11) | (5.21) |
| **Working Remotely x Post- lockdown** | 14.78*** | 20.04** | 12.20** | 267.06* | 446.42** | 52.52 † | -34.87*** | -20.61* | -49.34*** |
| | (4.83) | (8.81) | (5.78) | (148.84) | (226.18) | (185.72) | (9.49) | (11.12) | (15.15) |
| Working Remotely (1 yes 0 no) | -1.43 | -2.35 | -0.86 | -265.18** | -393.37** | -53.75 † | 5.67 | -5.05 | 17.44*,† |
| | (3.55) | (5.48) | (4.43) | (103.81) | (186.13) | (119.01) | (5.38) | (7.30) | (9.11) |
| Lockdown | 9.37*** | 10.91*** | 8.76*** | -116.93 | -182.50* | -17.30 † | -5.27 | -12.53** | 1.05 † |
| | (2.53) | (3.89) | (2.43) | (75.96) | (106.38) | (91.15) | (4.01) | (5.04) | (5.49) |
| Post-lockdown | 7.89*** | 9.35** | 6.42** | 135.76 | 150.04 | 132.54 | 4.36 | -4.11 | 12.85*,† |
| | (2.81) | (4.06) | (3.11) | (90.31) | (128.67) | (105.55) | (5.66) | (6.51) | (7.63) |
| Individual FE, Occupation FE, Time FE | Yes | Yes | Yes | Yes | Yes | Yes | Yes | Yes | Yes |
| Occupation FE x Time FE | Yes | Yes | Yes | Yes | Yes | Yes | Yes | Yes | Yes |
| Control variables | Yes | Yes | Yes | Yes | Yes | Yes | Yes | Yes | Yes |
| Mean Dependent Variable | 35.43 | 37.37 | 33.24 | 4900.88 | 5488.48 | 4227.20 | 172.83 | 180.36 | 164.19 |
| N | 4308 | 2301 | 2007 | 4308 | 2301 | 2007 | 4308 | 2301 | 2007 |

Notes: Remote is a dichotomous variable for whether the respondent worked fully from home in May 2020 during the COVID-19 lockdown. The reference time period is 'Pre-lockdown', between April-July 2018 and December 2019, prior to the COVID-19 pandemic. 'Lockdown' refers to March and June 2020, while 'Post-lockdown' refers to November 2020, six months after the end of the lockdown.

†+ denotes that the male-female differences in the estimated coefficients are statistically significant (p<0.05), based on the Seemingly Unrelated Estimation test. Standard errors, shown in the parentheses, are clustered at the household level.

*p<0.1

**p<0.05

***p<0.01

idiosyncrasies, occupation-specific trends, and occupation-specific time trends (hereafter called, 'occupation- and wave-specific confounding').

The coefficients of the standalone "remote work" variable—estimated with individual fixed-effects— capture the within-person average effect of adopting WFH across all time periods. On average, workers who adopted full WFH arrangements earned lower monthly income—by SGD $393.37 for men and SGD $53.75 for women—relative to when they were not working remotely, net of occupation- and wave-specific confounding. For male WFH workers, the lower monthly income came primarily from their earning lower hourly wages (by 2.35/hr, not significant, column 2) and working fewer monthly hours (by 5.05 hours per month, not significant, column 8). For female WFH workers, their lower monthly income was observed despite their working more monthly hours by 17.44/month (column 9)—weakly driven by their earning lower hourly wages by SGD $0.86/hr (column 3).

Turning to the total effect of WFH on monthly income in 2020 (columns 4–6), male WFH workers experienced a small loss in monthly income during lockdown (SGD -$5.94) but a gain in the post-lockdown period (SGD $53.05). Female WFH workers, by contrast, gained monthly income during lockdown (SGD $12.22) but experienced a slight decline post-lockdown (SGD -$1.23). Compared to non-WFH men, whose income dropped by SGD $182.50 during lockdown and rose by SGD $150.04 post-lockdown, male WFH workers fared better during the lockdown but saw smaller gains in post-lockdown. A similar contrast is seen among women: Female WFH workers gained income during lockdown, while their non-WFH counterparts lost income (SGD –$17.30); however, in the post-lockdown period, female non-WFH workers saw large gains (SGD $132.54), unlike female WFH workers who incurred a loss. Regarding the total effect of WFH on hourly wages in 2020 (columns 1–3), both male and female WFH workers experienced wage gains during lockdown and post-lockdown (SGD $4.89/hr and SGD $17.69/hr for men; SGD $7.87/hr and SGD $11.34/hr for women). While these gains were smaller than those of non-WFH workers during lockdown (SGD $10.91/hr for men, SGD $8.76/hr for women), they exceeded non-WFH counterparts' gains post-lockdown (SGD $9.35/hr for men, SGD $6.42/hr for women). In terms of the total effect on monthly work hours, the hours declined for both WFH and non-WFH males in 2020, but the decline was steeper among WFH males post-lockdown (–12.04 hours/month during lockdown and –25.66 hours/month post-lockdown) compared to non-WFH males (–12.53 hours/month and –4.11 hours/month, respectively) (columns 7–9). Among women, monthly work hours increased for both WFH and non-WFH groups during lockdown (4.46 hours/month for WFH, 1.05 hours/month for non-WFH), but diverged post-lockdown—where WFH women saw a sharp decline (–31.90 hours/month), while non-WFH women saw a continued increase (12.85 hours/month).

From the estimated coefficient for the interaction terms (i.e., the average treatment effect on the treated (ATT)), we observe several statistically significant and differential effect of the lockdown on WFH and non-WFH workers. More specifically, we see that the male WFH workers earned greater monthly salary income by SGD $387.43 during the lockdown and by SGD $446.42 in the post-lockdown period, compared to the male non-WFH workers in the respective periods—net of fixed individual traits and occupation- and wave-specific confounding (column 5). For women, no significant income differences were found between WFH and non-WFH groups during these periods (column 6).These gender differences were statistically significant.

The higher monthly income among the male WFH workers seemed to stem from their receiving higher hourly wages by SGD $7.24/hr during lockdown, and SGD $20.04/hr during post-lockdown (column 2), and *not* due to longer monthly work hours. In fact, male WFH individuals worked fewer monthly hours in 2020 (column 8). Among female WFH workers, their monthly income in lockdown/post-lockdown was unchanged because, in spite of their incurring higher hourly wages by SGD $8.73/hr in lockdown and SGD $12.20/hr in post-lockdown (column 3), they worked considerably fewer monthly hours—by SGD $12.98 hours/month in lockdown and SGD $49.34 hours/month in post-lockdown (column 9). In contrast, neither male non-WFH workers nor female non-WFH workers earned higher monthly income in 2020—as shown in the coefficients of "lockdown" and "post-lockdown" variables (columns 4–6). In fact, under lockdown, the non-WFH male workers earned lower monthly income and worked fewer monthly hours. The non-WFH males' and females' higher hourly

wages (SGD $9.35/hr for men, SGD $6.42/hr for women, columns 2–3) or non-WFH females' longer monthly work hours (by 12.85 hours/month, column 9) during the same period did not lead to substantial income gains. All estimates are net of fixed individual traits and occupation- and wave-specific confounding.

As a robustness check, we re-estimated the DID model without occupation and time fixed effects or occupation-specific time trends. Results shown in S13 Table confirm that the main findings (Table 4) are robust to the inclusion or exclusion of the occupation- and time- fixed effects in both direction and significance.

In a sensitivity check, we explored whether monthly income, hourly wages, and monthly work hours increased monotonically with the degree of WFH adoption—by replacing the binary WFH identifier in the difference-in-differences estimations with categorical indicators for five levels of WFH: fully from home, mostly from home, half-and-half, mostly outside, and fully outside. Results are shown in **Table 5**. From the coefficients of the "work arrangement" indicators, we observe that men who worked fully from home, and women who worked mostly or fully from home, earned lower monthly income across all periods compared to when they were not working remotely, net of wave-specific idiosyncrasies, occupation-specific trends, and occupation-specific time trends (columns 4–6). This pattern is consistent with Table 4. However, neither hourly wage (columns 1–3) nor monthly hours worked (columns 7–9) differed significantly by WFH intensity for both males and females across all periods.

The interaction terms indicate that, on average, the full WFH workers in both genders earned substantially higher monthly income, compared to the non-WFH counterparts in the respective periods, net of all controls and fixed-effects—suggesting that the effect of lockdown was statistically different for the two type of workers. Among women, only female partial-WFH workers saw similar gains (column 4–6). The higher monthly income among the female full/partial WFH workers was driven by their earning higher hourly wages (column 3). Yet, male full/partial WFH workers in lockdown did not earn a significantly higher hourly wages. For male full WFH workers, their higher monthly income was driven instead by their working fewer monthly hours in lockdown. Interestingly, among women, those working mostly outside also earned higher monthly income during the lockdown (column 6). This can be explained by their having the highest average monthly work hours (S6 Table), followed by those working doing WFH. In the post-lockdown period, the labor market outcomes were noticeably different for full WFH males and females. Overall, we did *not* find a monotonic progression of monthly income, hourly wage, or monthly work hours by the extent of WFH adoptions. Rather, the effects of lockdown/post-lockdown on labor market outcomes were explicitly seen among the full WFH males and females. While partial WFH respondents did earn higher monthly income and hourly wages during lockdown, these gains were not sustained—disappearing by November 2020. All results are net of fixed individual traits and occupation- and wave-specific confounding.

### 5.3. Mediating roles of household responsibilities and resources

Next, we explored the potential mediating roles of household responsibilities by conducting triple-differences (DDD) estimations. Our main regressors were the three-way interaction variables between remote work status, time, and mediators (childcare/chore times, availability of housework help and WFH spouses). Each mediator was examined in separate regressions due to high collinearity. Results are displayed in **Table 6**. The standalone components of the 3-way interaction were included in the estimations but were not shown (see S12 Table for further details). Because of the inclusion of the fixed-effects, our model estimated the mediating effects *net of* time-invariant individual traits as well as the occupation- and wave-specific confounding.

For the male WFH workers who incurred an income gain during the lockdown/post-lockdown periods (Table 4), their monthly income was not substantially mediated by the extent of childcare (column 5, panel A) or chores (column 5, panel B) they took on. However, for female WFH workers whose income gains in 2020 were not significant (Table 4), their monthly income decreased substantially if they carried out more chores: Spending one more minute every hour in chores for female WFH workers lead to a decrease in monthly income by SGD $88.26 during lockdown, and further by SGD $125.98 during post-lockdown (column 6, panel B). The result seems to have been driven by female WFH workers'

**Table 5. Effects of Remote Work Arrangements (Categorical) on Monthly Income.**

| | (1) | (2) | (3) | (4) | (5) | (6) | (7) | (8) | (9) |
|---|---|---|---|---|---|---|---|---|---|
| | Y = Hourly Wage | | | Y = Monthly Income | | | Y = Monthly Hours Worked | | |
| | Total | Male | Female | Total | Male | Female | Total | Male | Female |
| *(Reference = Pre-Lockdown, Working Fully Outside)* | | | | | | | | | |
| **Mostly outside x Lockdown** | -1.63 | -3.11 | 0.03 | 257.38 | 162.88 | 383.03* | -0.05 | -9.78 | 19.07 † |
| | (4.64) | (6.53) | (5.38) | (158.81) | (206.30) | (227.90) | (8.12) | (9.95) | (12.33) |
| **Half from home x Lockdown** | -0.93 | 0.06 | -1.55 | 238.17 | 324.38 | 133.41 | -3.90 | -4.83 | -2.95 |
| | (4.10) | (5.47) | (3.77) | (156.51) | (226.99) | (195.62) | (6.46) | (9.26) | (8.58) |
| **Mostly from home x Lockdown** | 8.27* | 3.74 | 15.15*** | 283.41* | 123.78 | 396.26** | -6.23 | -12.56 | -1.68 |
| | (4.96) | (6.75) | (5.27) | (165.46) | (219.39) | (197.54) | (7.07) | (8.44) | (10.24) |
| **Working remotely (Fully from home) x Lockdown** | 10.31** | 7.69 | 12.90** | 454.96*** | 514.40** | 292.89* | -12.50** | -12.75* | -12.01* |
| | (4.51) | (5.72) | (5.17) | (145.58) | (208.68) | (165.65) | (5.26) | (6.62) | (6.31) |
| **Mostly outside x Post-lockdown** | 0.97 | 6.79 | -11.51 † | 100.38 | 49.71 | 196.37 | -13.15 | -23.39 | 9.48 |
| | (6.31) | (8.56) | (7.68) | (178.41) | (240.91) | (255.83) | (14.85) | (16.90) | (25.87) |
| **Half from home x Post-lockdown** | 0.46 | -0.07 | 0.30 | -83.72 | -96.81 | -34.46 | -8.36 | -12.38 | -5.18 |
| | (5.22) | (6.79) | (5.55) | (170.69) | (262.19) | (212.08) | (13.59) | (18.83) | (14.34) |
| **Mostly from home x Post-lockdown** | 4.58 | 2.13 | 8.11 | -36.11 | -251.52 | 183.94 | -15.85 | -26.95* | -6.47 |
| | (5.33) | (7.89) | (5.17) | (191.52) | (275.41) | (243.54) | (12.18) | (15.49) | (16.98) |
| **Working remotely (Fully from home) x Post-lockdown** | 14.09** | 19.75** | 10.51* | 286.17* | 397.54* | 164.46 † | -40.39*** | -30.21** | -49.55*** |
| | (5.85) | (9.51) | (5.74) | (168.02) | (240.48) | (213.30) | (10.02) | (14.83) | (15.72) |
| *Lockdown Policy (Reference = Pre-Lockdown)* | | | | | | | | | |
| Lockdown | 7.76* | 10.63* | 4.65 | -293.42** | -306.25** | -234.88 | -2.60 | -7.24 | 1.06 |
| | (4.24) | (6.03) | (3.21) | (118.04) | (151.05) | (154.17) | (5.08) | (6.53) | (6.09) |
| Post-lockdown | 8.39* | 9.47 | 7.61 | 128.05 | 206.43 | 36.21 | 10.05 | 5.67 | 14.54 |
| | (4.96) | (6.66) | (4.84) | (114.36) | (154.40) | (156.61) | (8.24) | (11.20) | (9.16) |
| *Work Arrangements(Reference = Work Fully Outside)* | | | | | | | | | |
| Mostly outside | 7.11 | 6.00 | 11.03 | -125.64 | 25.13 | -310.09 | 3.16 | 10.14 | -9.98 |
| | (5.65) | (7.13) | (8.91) | (164.50) | (227.78) | (224.28) | (12.29) | (15.62) | (18.82) |
| Half from home | 8.25 | 7.35 | 9.54 | -101.96 | -39.64 | -196.98 | 3.10 | -3.96 | 15.61 |
| | (6.15) | (6.41) | (11.29) | (157.65) | (226.64) | (205.11) | (10.11) | (14.67) | (13.65) |
| Mostly from home | 10.42 | 12.91* | 6.31 | -199.85 | 34.87 | -412.33**,† | -7.70 | -0.72 | -6.89 |
| | (6.94) | (7.40) | (12.90) | (177.85) | (255.02) | (208.64) | (10.70) | (13.77) | (16.92) |
| Working remotely (Fully from home) | 9.26 | 7.59 | 10.43 | -412.70*** | -416.97* | -336.78** | -0.57 | -10.73 | 15.17 |
| | (7.11) | (8.00) | (12.51) | (143.50) | (220.98) | (171.02) | (8.35) | (11.53) | (12.69) |
| Individual FE, Occupation FE, Time FE | Yes | Yes | Yes | Yes | Yes | Yes | Yes | Yes | Yes |
| Occupation FE x Time FE | Yes | Yes | Yes | Yes | Yes | Yes | Yes | Yes | Yes |
| Control variables | Yes | Yes | Yes | Yes | Yes | Yes | Yes | Yes | Yes |
| N | 4308 | 2301 | 2007 | 4308 | 2301 | 2007 | 4308 | 2301 | 2007 |

Notes: Time period specifications are the same as in the previous tables.

†+ denotes that the male-female differences in the estimated coefficients are statistically significant (p < 0.05) based on the Seemingly Unrelated Estimation test. Standard errors, shown in the parentheses, are clustered at the household level.

*p < 0.1

**p < 0.05

***p < 0.01

**Table 6. Mediating Role of Household Responsibilities on Hourly Wage, Monthly Income, and Monthly Hours Worked.**

| | (1) | (2) | (3) | (4) | (5) | (6) | (7) | (8) | (9) |
|---|---|---|---|---|---|---|---|---|---|
| | Y = Hourly Wage | | | Y = Monthly Income | | | Y = Monthly Hours Worked | | |
| | All | Male | Female | All | Male | Female | All | Male | Female |
| *Panel A: Role of Childcare (minutes/hr)* | | | | | | | | | |
| **Remote x Lockdown x Childcare** | -0.16 | -0.75 | 0.13 | -16.44 | 25.95 | -17.92 | 0.70 | 0.82 | 0.95* |
| | (0.93) | (0.80) | (1.17) | (14.18) | (30.97) | (15.90) | (0.47) | (1.15) | (0.52) |
| **Remote x Post-Lockdown x Childcare** | -0.35 | 0.52 | -0.37 | -25.53* | 7.45 | -27.75 | 0.36 | -0.15 | 1.14 |
| | (0.94) | (1.35) | (1.22) | (17.16) | (32.13) | (20.26) | (0.93) | (1.58) | (1.16) |
| *Panel B: Role of Chores (minutes/hr)* | | | | | | | | | |
| **Remote x Lockdown x Chores** | -0.58 | 0.92 | -2.28** | -78.52* | 22.48 | -88.26*,† | 0.01 | 0.03 | 2.39 |
| | (0.97) | (2.08) | (1.00) | (47.35) | (102.78) | (44.89) | (1.77) | (2.64) | (2.26) |
| **Remote x Post-Lockdown x Chores** | -1.63 | 2.72 | -4.14** | -125.99** | -44.03 | -125.98**,† | -4.41 | -2.57 | -0.24 |
| | (2.62) | (8.08) | (1.82) | (61.89) | (127.14) | (58.13) | (5.94) | (7.39) | (7.62) |
| *Panel C: Role of Whether Respondents Receive Any Help for Household Work (1 yes 0 no)* | | | | | | | | | |
| **Remote x Lockdown x Receiving any help** | 11.72 | 5.48 | 18.11* | 319.80 | 537.84 | 229.92 | 0.03 | 10.56 | -12.43† |
| | (6.16) | (7.32) | (9.43) | (227.79) | (363.85) | (257.60) | (8.01) | (10.50) | (11.05) |
| **Remote x Post-Lockdown x Receiving any help** | 2.05 | -13.57 | 15.06 | 581.18* | 574.47 | 644.64* | 30.12 | 18.04 | 37.85 |
| | (10.63) | (19.90) | (9.72) | (308.33) | (476.32) | (379.75) | (23.81) | (23.61) | (40.53) |
| *Panel D: Role of Whether Respondents have Remotely-Working Spouses (1 yes 0 no)* | | | | | | | | | |
| **Remote x Lockdown x Spouse working remotely** | 14.40** | 9.12 | 15.70* | 446.46** | 706.87** | 167.06† | 1.95 | 11.34 | -4.47† |
| | (6.03) | (6.36) | (8.55) | (223.62) | (351.66) | (273.34) | (7.91) | (9.66) | (10.73) |
| **Remote x Post-Lockdown x Spouse working remotely** | 22.66* | 29.35* | 11.55 | 579.22* | 990.47** | -18.33† | 19.18 | 8.77 | 9.29 |
| | (12.49) | (18.36) | (12.36) | (308.11) | (497.07) | (318.97) | (29.69) | (27.62) | (35.49) |
| Individual FE, Occupation, Time FEs | Yes | Yes | Yes | Yes | Yes | Yes | Yes | Yes | Yes |
| Occupation FE x Time FE | Yes | Yes | Yes | Yes | Yes | Yes | Yes | Yes | Yes |
| Control variables | Yes | Yes | Yes | Yes | Yes | Yes | Yes | Yes | Yes |
| N | 4308 | 2301 | 2007 | 4308 | 2301 | 2007 | 4308 | 2301 | 2007 |

Notes: Notes: Panels A-D are run separately because the indicators of household responsibilities are highly collinear. Each component of the interaction terms are also included in the regressions, but are not shown in the table. The reference time period is 'Pre-lockdown', between April-July 2018 and December 2019, prior to the COVID-19 pandemic. 'Lockdown' refers to March and June 2020, while 'Post-lockdown' refers to November 2020, six months after the end of the lockdown.

†+ denotes that the male-female differences in the estimated coefficients are statistically significant (p < 0.05) based on the Seemingly Unrelated Estimation test. Standard errors are clustered at the household level.

*p < 0.1

**p < 0.05

***p < 0.01

sizeable loss in hourly wages—by SGD $2.28/hr during lockdown and SGD $4.14/hr during post-lockdown—that corresponded to one minute increase in chores/hr (column 3, panel B).

Despite the insignificant income gain of female WFH workers shown in Table 4, their monthly income still increased significantly by SGD $644.64 when they received any help for housework (column 6, panel C). While their hourly wages in the same situation grew, it was not statistically significant (column 3, C). Compared to women, male WFH workers' monthly income or hourly wages did not change significantly by the availability of help (columns 2, 5, C). Instead, male WFH workers' monthly income grew substantially to SGD $990.47 by the availability of WFH spouses (column 5, panel D)—perhaps driven by the hourly wages increase by SGD $29.35/hr under the same circumstance (column 2, panel D).

Having a spouse who worked fully remotely did not help augment the income gains of female WFH workers: Albeit statistically insignificant, we observe a reduction in monthly income (column 6, panel D). Interestingly, the monthly work hours of both genders were un-affected by the extent of childcare/chores done, availability of help for housework, or spouse's WFH statuses (columns 7–9, panels A-D). We discuss the implications of all our findings in Section 6.

## 6. Discussion

Our difference-in-differences analysis reveals that the impact of remote work on labor market outcomes varied substantially by gender and work arrangements during and after the government-mandated lockdown in Singapore (i.e., a strict social distancing measure to fight COVID-19). Prior to the COVID-19 outbreaks in 2018–2019, WFH workers earned lower monthly income and hourly wages than their non-WFH counterparts—suggesting that individuals with full WFH arrangements may have sorted into remote jobs despite lower pay or less income security, possibly due to heavy housework responsibilities. In 2020, with the onset of the pandemic and the implementation of mandatory social distancing policies across many countries, the labor market outcomes of WFH jobs underwent notable changes.

Specifically, male workers who adopted full WFH arrangements experienced a modest income loss during the lockdown (March–June 2020), relative to non-WFH male workers. Although they earned higher hourly wages, a substantial decline in hours worked resulted in an overall income loss during this period. Their fate changed in post-lockdown (November 2020) as male WFH workers saw modest monthly income gains compared to non-WFH counterparts, driven by increased hourly wages that offset further reductions in work hours. Female workers who adopted full WFH arrangements experienced slight monthly income gains during the lockdown, compared to non-WFH females—driven by increases in both hourly wages and monthly work hours. However, their monthly income declined marginally in the post-lockdown period due to a significant reduction in work hours, which offset the rise in hourly wages. Overall, compared to non-WFH counterparts, both male and female WFH workers experienced smaller fluctuations in salary income during the lockdown but smaller income gains post-lockdown.

Our data partially support *Hypothesis 1* for male WFH workers, who experienced delayed income gains relative to non-WFH peers in the post-lockdown period—driven by productivity gains reflected in the higher hourly wages. However, the hypothesis is not supported for female WFH workers, whose monthly income did not significantly increase despite higher hourly wages, primarily due to reductions in work hours. The findings strongly support *Hypothesis 2*: Female WFH workers consistently experienced smaller income gains than their male counterparts, particularly in the post-lockdown period. Meanwhile, there is *limited evidence* that the gender gap narrowed in the post-lockdown period, as female WFH workers continued to face constraints on work hours and earnings, suggesting only partial convergence.

One may wonder about the large magnitude of the increase in hourly wage of WFH workers in post-lockdown period. For instance, male WFH workers' hourly wage rose by SGD $17.69/hour in post-lockdown (**Table 4**, column 2), when their pre-lockdown wage was SGD $34.95/hour (S11 Table, column 1). We confirm that the steep increase we observe in post-lockdown was not due to inflation in 2020. Instead, part of this increase is likely mechanical in nature. Male WFH workers experienced a reduction of 25.66 hours in monthly work hours while simultaneously earning higher monthly income in post-lockdown. These two outcomes mechanically induces higher hourly wages in that *we calculated the hourly wages using the monthly income and monthly hours worked*. Additionally, Singapore launched several COVID-19 relief policies that enabled full-time workers adopting WFH arrangements to *retain their jobs and monthly income* even if they could not work as many hours from home (i.e., due to loss of productivity, blurred boundaries between work and responsibilities at home). One such policy was Singapore's Job Support Scheme (JSS), which encouraged employers to retain employees by funding 75% of the salaries—capped at the first SGD $4,600—of Singaporeans and permanent residents during April and May 2020 and by launching a tier-based funding (75% for industries directly hit by COVID-19, 25–50% for the rest of the industries) for the remaining 15 months until March 2021 [73,74]. Furthermore, as the Singapore government encouraged job diversification during the pandemic [64,65], some WFH workers may have taken on part-time,

high-paying roles, which further increased monthly income despite reduced hours. These policies likely contributed to maintaining or increasing monthly income despite a reduction in work hours, thereby increasing the hourly wage.

It is important to note that the higher hourly wages associated with adoption of WFH arrangements for both genders in 2020 were not driven by changes in demand for remotely conducted occupations, as we controlled for occupation-specific time trends between 2018 and 2020. Thus, the changes in work environment and productivity appear to be the primary drivers of rising hourly wages. Although our study does not explore the precise mechanisms behind increased productivity, the literature suggests factors such as enhanced flexibility, fewer peer interruptions, and reduced commuting stress [33–35].

It is interesting to observe that female WFH workers reduced their monthly work hours although their hourly wages increased. In the post-lockdown period, female WFH workers' monthly work hours were lower by as much as 31.90 hours/month. A significant portion of this freed time appears to have been reallocated to childcare. An increase of 0.62 min/hr spent on childcare (Table S9, column 6) translates to approximately 14.88 min/day or 7.44 hrs/month—roughly one-quarter of the freed time that came from the reduction in work hours. Due to data limitations, we were unable to account for the remaining time, but anecdotal evidence suggests that WFH workers in Singapore bore additional administrative duties at home, such as completing daily health declarations for family members. Additional responsibilities for WFH workers in pandemic came from elderly care, given that approximately 55% of seniors aged 65 or older lived with their children in 2020 [75]. While the responsibilities for both activities heightened during the pandemic, such activities would not have been captured in our survey data.

The mediator assessments via triple-differences estimations indicate that unpaid household responsibilities played an important role, particularly for women. For female WFH workers, increased chore time was associated with notable declines in both monthly income and hourly wages. Conversely, access to external help was linked to higher income, although not necessarily higher wages. Male WFH workers benefited from having a spouse who also worked remotely, experiencing substantial increases in monthly income and hourly wages—a pattern not observed for female WFH workers. The availability of external help increased the monthly income of female WFH workers in the post-lockdown period, although this cannot be sufficiently explained by their increased hourly wages (statistically insignificant) or monthly work hours (insignificant). We speculate that the monthly income growth of female WFH workers receiving external help for housework may reflect non-wage compensation such as childcare subsidies introduced post-lockdown. Such introduction of new in-kind benefits would have been captured in the WFH mothers' monthly income but not hourly wages. The availability of fully remotely-working spouses increased monthly income of male WFH workers through a substantial increase in their hourly wages—a proxy for productivity—in the post-lockdown period, but generated no changes among female WFH workers. If anything, female WFH workers with WFH spouses experienced slight income reductions, although these were not statistically significant.

These findings support *Hypothesis 3,* reinforcing that household responsibilities disproportionately affected women who often served as primary caregivers for unpaid housework [49]; approximately 64–66% of female WFH workers in our data devoted above-median time in childcare/chores, compared to 39–40% of male WFH workers. The pandemic appears to have amplified preexisting gender disparities. Case in point, **Fig 3** shows that prior to the pandemic in 2018, both male WFH workers and female WFH workers spent more time on childcare and chores. The May 2020 lockdown led to spikes in time allocated to both childcare and chores among all groups, but especially for the female WFH workers in terms of childcare and both for female WFH and non-WFH workers in terms of chores (as can be seen by the slopes). This aligns with the literature showing more substantial shifts in time-use patterns from work to household and childcare duties among women relative to men during the pandemic [20,22]. Ultimately, heavier household responsibilities—measured through chore time—were linked to significantly lower income and wages for women, but not for men, underscoring the gendered impact of unpaid labor on productivity and earnings.

Some may question the intuitive nature of these mediation results. From the perspective of time allocation to work and household responsibilities, time spent on household production (e.g., chores, childcare) should substitute for work time,

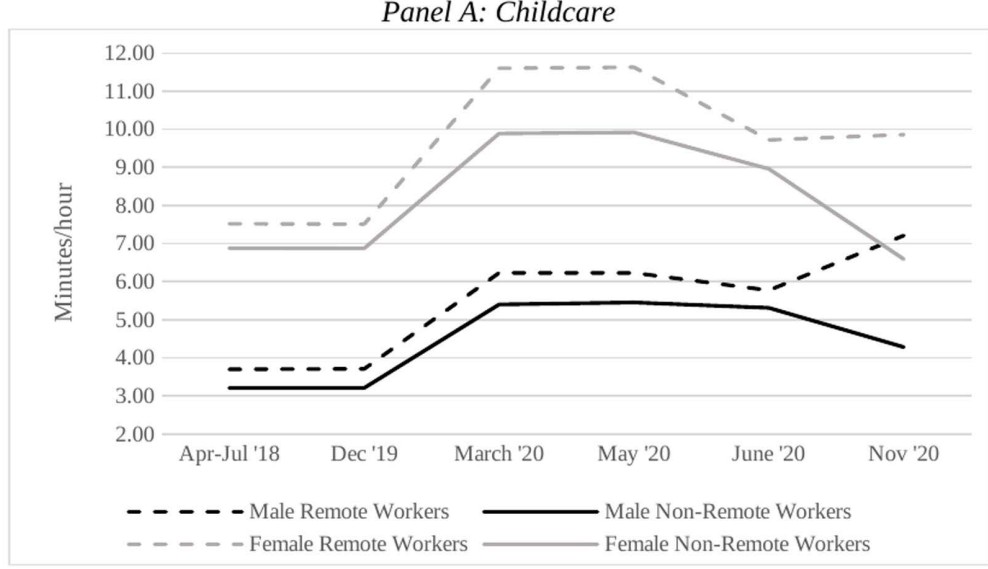

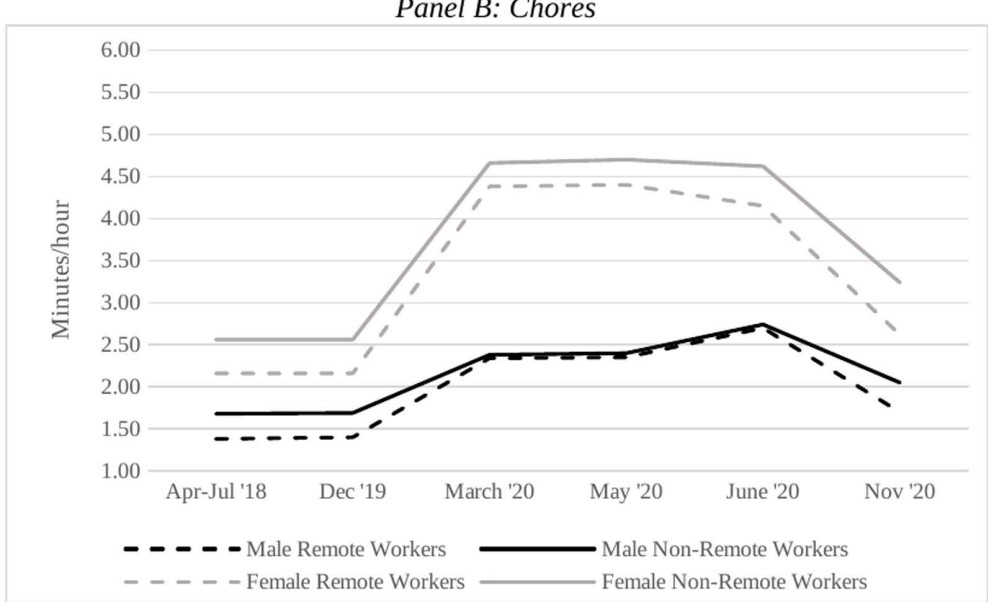

**Fig 3. Time Use on Childcare and Household Chores, 2018-2020.**

resulting in lower monthly income. However, this theory is grounded in conventional work arrangements outside the home. In our modern-day labor market settings with WFH arrangements in place, the boundaries between work and home are blurred. The substitutability between work and housework may not hold in this new context. For instance, it is conceivable that a parent may supervise a child while working (e.g., running codes or performing other remote tasks). These activities could take place simultaneously or even be complementary. On a related note, remote work may also increase total time available, as commuting and workplace socializing are reduced. Thus, increased household duties (e.g., childcare) may not directly reduce work time if newly freed time from commuting is reallocated on childcare.

Consistent with this reasoning, our mediation analysis found that childcare and chores did not significantly mediate work hours for either gender. We further examined this relationship using panel fixed effects and DID models with childcare and chores as outcome variables (S9 and S10 Tables), using the same covariates and fixed-effects as in main results (Table 4). Results show that WFH was positively associated with childcare and chore responsibilities for both genders, but more strongly for women. These findings suggest that WFH adoption imposed heavier domestic burdens, particularly on female workers. If WFH and childcare/chores were substitutes for mothers, the blurred boundary between home and work-space—facilitated by the adoption of WFH and increased chores/childcare duties—could have taken a toll on the mothers' labor market outcomes. These results further highlight a gendered divergence in the experience of WFH.

This study has several limitations. First, our sample is not fully nationally representative due to non-probabilistic sampling and attrition across survey waves. In comparing our sample's sociodemographic traits with those of the population shown in the Census of Population 2020 [76] and 2020 Labour Force in Singapore [8], we observe that the birth parities of respondents in our sample is comparable to national statistics. At the same time, our data oversamples respondents who are highly-educated, professional, and high-income. Additionally, by restricting the sample to continuously employed individuals throughout the survey waves, we likely underrepresent those who exited the labor force during 2020. This may have overestimated the labor market outcomes of full WFH workers in our study. The sample may also overrepresent households with young children, as indicated by a rise in the share of mothers with infants during the lockdown. Future research addressing the current data limitations would help strengthen our findings.

Second, although the nationwide lockdown in Singapore limited self-sorting into WFH and non-WFH roles, some residual sorting may remain, particularly through *switching between jobs.* We believe this is minimal, given the unexpected nature of the pandemic and movement restrictions. However, some sorting may explain observed increases in the share of workers in non-WFH roles post-lockdown. A related concern is that our final sample suffers from potential selection bias due to our sample restriction to continuously employed individuals across waves—for the reasons elaborated in Section 3.1. That said, only 8.3% of our sample (60 unique individuals) were dropped due to unemployment in 2020. Unlike in many other countries, COVID-19 induced unemployment was not a widely-observed phenomenon in Singapore, especially for the Singaporean nationals and permanent residents. This was due to government policies that ensured continued employment during the pandemic, such as the Job Support Scheme [73,74].

Third, our analysis does not include industry fixed effects due to data limitations. The concern is that the industries would indicate the essential or non-essential nature of work respondents carried out—which, in turn, had implications for their hourly wage rates as the essential workers experienced rising demands during the pandemic. However, we believe the bias from missing industry fixed-effects is minimal in our study. While certain industries such as healthcare included a large number of essential workers, not all jobs in the industries were essential. In other words, the essential/non-essential labels as well as any economic consequences of the labels are, in fact, *occupation-specific* rather than industry-specific. Indeed, studies show that industries are generally less predictive than occupations with regard to changes in labor market outcomes during the pandemic [2,39]. We accounted for the job-specificity by including occupational fixed effects in the estimations. More importantly, we accounted for changes in job demands between 2018–2020 (e.g., soaring demands for healthcare frontline workers during the COVID-19) by controlling for occupation-specific time trends in all estimations.

Fourth, our data do not differentiate between part-time and full-time status, as respondents reported total monthly work hours across all jobs. The monthly work hours are not an accurate indicator of part-time/full-time statuses as respondents in our survey reported the total monthly work hours from all jobs and the government of Singapore encouraged individuals to seek multiple jobs during the pandemic to enhance their economic security [64,65]. In an effort to further elucidate how the COVID-19 induced changes affected vulnerable worker groups vis-à-vis the adoption of remote work, future research should examine the welfare of part-time workers.

Lastly, as our data was collected from female spouses, the records we have on male spouses are based on indirect reporting. While this could have introduced a degree of reporting errors, we do not conclude that the information regarding

the male spouses is systematically less accurate than the information on female respondents in our data. When all responses are self-reported, the directionality or the magnitude of the reporting errors for respondents and their spouses are unclear. Existing literature suggests that the respondents tend to understate their spouses' contributions while over-stating their own [77,78], while overestimating spouses' negative conditions (e.g., pain due to illnesses) [79]. The extent of accuracy, over-reporting, or under-reporting of spousal responses also depend on the nature of spousal relationships—for instance, whether there are similarities in situation across spouses [80]. In fact, several studies report a strong evidence of accuracy of spousal responses and convergence across spouses [80,81]. Regardless, we would benefit from future research that re-evaluates our findings using direct responses from both male and female spouses.

In spite of these limitations, our study provides valuable insights on the effects of the adoption of remote work arrange-ments on male and female spouses' labor market outcomes. Our detailed household-level data on background character-istics, including household composition and presence of helpers, as well as both spouses' time use on household chores and childcare also yields insights into how underlying gendered differences in domestic responsibilities may result in dif-ferential gains from WFH arrangements across genders. Importantly, we contribute evidence from a non-Western, smaller economy, adding diversity to the existing literature.

## 7. Conclusion

Our findings highlight that the benefits of remote work during the pandemic were not distributed evenly. While male WFH workers captured sustained gains, female WFH workers' gains were limited by disproportionately heavier household labor burdens, which in turn led to constrained work hours. These findings carry important policy implications, particularly as remote work becomes a semi-permanent feature of the labor market. To ensure more equitable distribution of benefits and inclusive recovery across genders, policymakers should look beyond simply encouraging or mandating flexible working arrangements. Addressing gender inequality in household labor is essential. To this end, wider recognition is needed that household responsibilities can diminish the benefits of flexible work arrangements. Workplace initiatives should address persistent negative attitudes and discrimination towards men who seek to take advantage of flexible working arrange-ments to increase paternal involvement in childcare/household duties. These cultural norms discourage men's participa-tion in housework, reinforcing gendered labor divisions [82]. Hence, post-pandemic labor policy should encompass not only offering remote work options but also making efforts to balance their uptake across genders and reduce discrimina-tory attitudes and norms that perpetuate unequal division of household responsibilities.

## Supporting information

**S1 Fig. Changes in Remote Work Arrangements over the Analytic Period.**
(DOCX)

**S1 Text: Labor Market in Singapore.**
(DOCX)

**S2 Text: Sensitivity Analyses - Using a Broader Definition of the Remote Work Indicator.**
(DOCX)

**S3 Text: Exploring the Links between WFH and Childcare/Chores.**
(DOCX)

**S1 Table. Employed Singaporean Residents Aged 15+ by Occupation and Gender, 2010–2020 (in 000).**
(DOCX)

**S2 Table. Employed Singaporean Residents Aged 15+ by Industry and Gender, 2020 (in 000).**
(DOCX)

**S3 Table.  Occupational Categories in the Survey.**
(DOCX)

**S4 Table.  Remote Work Preferences Among Female Respondents in November 2020.**
(DOCX)

**S5 Table.  Comparison of Baseline Characteristics in the Sample and National Statistics of Married Resident Mothers Aged 25–59.**
(DOCX)

**S6 Table.  Average Working Hours by Month and Remote Work Arrangements.**
(DOCX)

**S7 Table.  Assessment of Pre-Trends: Monthly Income and Monthly Hours Worked.**
(DOCX)

**S8 Table.  Effects of Remote Work Arrangements on Hourly Wage, Monthly Income, and Monthly Work Hours.**
(DOCX)

**S9 Table.  Effect of Remote Work Arrangements (Categorical) on Time Spent on Childcare.**
(DOCX)

**S10 Table.  Effect of Remote Work Arrangements (Categorical) on Time Spent on Chores.**
(DOCX)

**S11 Table.  Descriptive Statistics for Labor Market Outcomes by Detailed WFH Categories.**
(DOCX)

**S12 Table.  Mediating Role of Household Responsibilities on Hourly Wage, Monthly Income, and Monthly Hours Worked, Detailed.**
(DOCX)

**S13 Table.  Effects of Remote Work Arrangements on Hourly Wage, Monthly Income, and Monthly Work Hours**
(DOCX)

## Author contributions

**Conceptualization:** Zeewan Lee, Poh Lin Tan, Jie-Sheng Tan-Soo.

**Data curation:** Poh Lin Tan.

**Formal analysis:** Zeewan Lee.

**Funding acquisition:** Poh Lin Tan.

**Investigation:** Zeewan Lee, Poh Lin Tan, Jie-Sheng Tan-Soo.

**Methodology:** Zeewan Lee, Jie-Sheng Tan-Soo.

**Resources:** Poh Lin Tan.

**Software:** Zeewan Lee.

**Validation:** Zeewan Lee.

**Visualization:** Zeewan Lee.

**Writing – original draft:** Zeewan Lee, Poh Lin Tan.

**Writing – review & editing:** Zeewan Lee, Poh Lin Tan, Jie-Sheng Tan-Soo.

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
