## [Decision Letter · Decision Letter 0]

20 Mar 2024

PONE-D-24-02155Unequal Gains from Remote Work during COVID-19 between Spouses: Evidence from Longitudinal Data in SingaporePLOS ONE

Dear Dr. Lee,

Thank you for submitting your manuscript to PLOS ONE. After careful consideration, we feel that it has merit but does not fully meet PLOS ONE’s publication criteria as it currently stands. Therefore, we invite you to submit a revised version of the manuscript that addresses the points raised during the review process.

We look forward to receiving your revised manuscript.

Kind regards,

José Alberto Molina

Academic Editor

PLOS ONE

Journal Requirements:

This project was supported by funding from the Lee Kuan Yew School of Public Policy, National University of Singapore [R-603-000-347-115; A-0003976-00-00; R-603-000-237-133]. 

3. In the online submission form, you indicated that The raw data for this study cannot be made publicly available due to restrictions on use of collected data under the IRB protocol. Aggregated data will be available (contact Corresponding Author via email: zeewan.lee@nus.edu.sg) for interested researchers.

Reviewers' comments:

Reviewer's Responses to Questions

**Comments to the Author**

1. Is the manuscript technically sound, and do the data support the conclusions?

Reviewer #1: Yes

Reviewer #2: Partly

2. Has the statistical analysis been performed appropriately and rigorously? 

Reviewer #1: Yes

Reviewer #2: No

3. Have the authors made all data underlying the findings in their manuscript fully available?

Reviewer #1: Yes

Reviewer #2: No

4. Is the manuscript presented in an intelligible fashion and written in standard English?

Reviewer #1: Yes

Reviewer #2: Yes

5. Review Comments to the Author

Reviewer #1: The paper examines whether the gains and losses from adopting WFH on monthly income vary by gender using the mandated lockdown as an exogenous shock. Employing the difference in differences estimator, the authors find that the lockdown increases earned income for men who work from home, but not for women. They argue that these gender differences may come from the differential childcare and housework burden.

Major Points

- I was wondering if data contain information on hours worked because it is crucial when we interpret the results.

- With this information, we can understand whether income gains come from an increase in wage rate or in hours worked. At the end of section 2.1, Hypothesis 1 seems to be based on the idea that WFH improves productivity, which should have positive impacts on income through wages. However, the story would be different if it comes from a change in hours worked, which may be more consistent with the story about the within-household division of housework. These should be clearly mentioned and investigated if data is available.

- The results in section 5.2 state that WFH increases income for men but not for women relative to non-remote workers. However, based on the observation in Figure 2, income trajectories for teleworkers are not different by gender, at least in the short run. What is different is the one for non-teleworkers – female non-teleworkers experience less income loss in the short run after the lockdown. This, again, matters in interpreting the result because this result could suggest that the impact of WFH may be the same by gender, while the impact of the lockdown and/or the pandemic on non-teleworkers may be different by gender. For example, male non-teleworkers might have to reduce their hours worked more than women due to some characteristics of jobs. If this is true, then the argument might be “wages for non-teleworkers do not decline for women but do for men,” which is unrelated to the effectiveness of WFH. The current analysis cannot rule out this possibility.

- Still, it might be possible to state that the notion of the “effect of WFH” includes the above-mentioned possibility. Even in this case, this possibility should be mentioned, and how the readers can understand the effect of WFH in this scenario should be given so that they can understand the policy implications in a more coherent way.

- In the analysis in section 5.3, the potential mediating roles of household responsibilities are discussed. What is the definition of the time use variable in this analysis? Namely, is this time spent on childcare at time t, or the pre-lockdown level? This is an important difference because if the former is true, then the effect seems mechanical because increasing time spent on childcare almost mechanically reduces the hours worked, which in turn decreases income. The reason why the result is insignificant for men then may be because men don’t spend time on housework. I would suggest that the authors may want to check the effects of WFH on time spent on childcare and housework in separate regressions with the same specification, which would make these stories clear. Again, the point is distinguishing between “productivity decline” and “reduced hours worked” mentioned above.

Reviewer #2: In this paper the authors conducted an ad hoc survey targeting married women in Singapore, for whom they collect longitudinal information at 6 points in time (from 2018 to November 2020) on both themselves and their male partners. The aim is to analyse the effect of the COVID-19 pandemic and the subsequent period on monthly wages and to see if there have been differences in the effects of the pandemic between teleworkers and non-teleworkers across genders. The authors found a significant salary income growth among remotely-working male spouses, but not among female spouses.

The paper addresses a relevant topic related to the effects of the COVID-19 pandemic on wages and gender inequality. It is an interesting paper, and it is, overall, well written.

I have several comments related to the organisation of the paper and the need for clarification of some aspects. The motivation in the introduction should also be improved. There are aspects that in my opinion are key to properly motivate the article (such as telework figures in Singapore, the effects of telework on productivity, etc. ....) that should be explained in the Introduction (see my specific comments below).

Despite having a general favourable opinion of the paper, my main concern lies with the estimation results. Firstly, the authors analyse monthly salaries, but they do not control for the number of working hours despite having information on this issue. Secondly, it is no clear if the analysed individuals are employed consistently across all periods. For me, it would be surprising that all individuals and their partners were consistently employed throughout. Third, the definition of teleworkers is very rigid (only those workers who fully work from home), which implies that their main explanatory variable (working from home) is almost zero before the pandemic. Lastly, there are some inconsistencies between the coefficients obtained in the estimations for the total sample (pooled) and the separate estimations for men and women (see for example the coefficient of the variable "lockdown" in the three samples in Table 4). The coefficient is negative for the pooled sample (implying that during the lockdown the monthly wages of non-teleworkers decreased). However, when separate estimates are made for men and women, the coefficient is positive for both, indicating just the opposite. There can also be observed several inconsistencies for other coefficients. For these reasons, I believe that the authors should carefully review the regression results. I therefore consider that the article requires major revisions.

MAIN COMMENTS (ORGANIZED FOLLOWING THE MANUSCRIPT STRUCTURE):

INTRODUCTION

1) In the introduction I miss some figures (based on official statistics) on telework in Singapore before and after the pandemic, differentiating by gender.

2) Although explained on page 14 of the manuscript, it would be useful to mention also in the introduction that the authors exploit a non-official ad-hoc survey approved by the National University of Singapore Institutional Review Board. Additionally, the purpose of the survey should be stated.

3) In the introduction, it should be specified that the salary income is measured on a monthly basis. I guess that the observed increase 'in monthly salary' reported by the authors is related to an increase in the number of working hours (that authors do not explain nor mention).

4) On page 4, last paragraph, the authors state, as an additional contribution of their article, the fact that “The current literature on the labor market impact of Covid19 primarily relies on evidence from Western or larger economies. Such evidence may not apply to non-Western smaller economies given the international heterogeneity in the institutional responses to Covid19, extents of adoption of remote work, and sociocultural contexts”. I suggest adding a short sentence highlighting the main differences between Western and non-Western countries in the fight against COVID-19.

5) Figure 1 is mentioned on page 5, last paragraph, and the authors say that by the fourth quarter of 2020 the economy had almost bounced back to pre-pandemic activity levels (“By the last quarter of 2020, the economy had almost bounced back to pre-pandemic activity levels, which coincided with a steep decline in number of daily confirmed cases to mostly single digits (see Figure 1)”. However, looking at Figure 1, it appears that the growth rate of GDP in the fourth quarter of 2020 is negative (could it be a visual effect on the horizontal axis?). I assume that the growth rates shown in Figure 1 are year-on-year growth rates, but it would be useful to make this explicit in the figure. In addition, it would be useful to add employment growth, differentiated by gender.

SECTION 2 – BACKGROUND

6) I find it a bit confusing to mix the background section, which is usually a literature review, with aspects of the specific contribution of the article as well as the hypothesis statement of the paper. I suggest revising this section so that it is more standard with what is usually done in academic articles. A simple option would be to change the title of Section 2. For me, it would be more natural (it is a personal taste) to focus the background section on literature review and specify the hypotheses in the Estimation section, before the equation 1, to better link how the three hypotheses of the article will be tested through equation 1.

7) On page 8, the authors say “In case of Singapore, based on the qualitative evidence that the share of respondents who reported productivity increases following the introduction of remote work rose from 15.1% in April 2020 to 22.5% in June 2020, and the share of respondents who were concerned about productivity declines fell from 78% to 40%”. In my opinion this should be in the Introduction to support the motivation of the article. Also, in relation to Hypothesis 1, the authors should explain the mechanism by which they assume that teleworking implies an improvement in wages. This should be explained more clearly.

8) On page 9, first paragraph, the labour market in Singapore is partially discussed. Personally, I do not know the characteristics of the labour market in Singapore. I think that for the international reader, especially those from western countries, it would be very useful to provide some figures on the labour market from a gender perspective in Singapore. What is the employment rate of men and women? How is employment distributed between full-time and part-time between men and women? It would also be useful to show a table, even if only in the appendix, with the distribution of employment by occupations and industries differentiated by gender, in order to see which occupations and industries are more feminised in the case of Singapore.

9) As the authors partially mention, women were more likely to leave employment or reduce hours worked than men. This is a phenomenon that has been noted especially for the literature based on western countries. Did this happen in Singapore as well ? Can the authors provide some figures to contextualise this phenomenon for Singapore? How did the fall in GDP affect the employment of both men and women?

10) A personal taste: I find the transition to section 3 a bit rough. Perhaps it would be appropriate to end with a few brief lines to give way more naturally to the data section.

SECTION 3 – DATA AND METHODS

11) When explaining the database, it is necessary to specify that the survey is based on ad-hoc survey (I imagine that was conducted by the researchers in the context of a research project endorsed by the National University of Singapore). Additionally, it is necessary to specify the purpose of the survey. Given that the survey started in 2018 it was impossible to know at that time that there was going to be a pandemic in 2020. What was the original objective of the survey?

12) It is important to describe briefly the type of information collected in the survey.

13) As the face-to-face surveys in 2018 were conducted on the street and only among women indicates (I imagine) that there is a proportion of female respondents who are NOT working. Were the surveys conducted during "non-working" hours? The authors state that the criteria for being surveyed are "Participants met the following inclusion criteria: currently married in 2018; either a Singaporean citizen or married to a Singaporean citizen, and able to read, write and speak in English". Therefore, being employed is not included as a criterion. Given that the survey collects wage information, were all female respondents employed in 2018? How is the distribution of the sample in terms of percentage of employment compared to what is observed in Singapore's official statistics? Were the husbands also employed?

14) In Table 1, it is clear that the first interview was face-to-face, but the verbal explanation on page 12, last paragraph, should also be stated.

15) I consider that the elaboration of an ad-hoc survey is in itself a contribution of the article and the fact that the authors conduct a survey is because the type of information collected in official surveys in Singapore is not sufficient for the purposes of the article. I do not know the characteristics of official surveys in Singapore, but at least in European countries the type of information collected by the authors is included in labour force surveys and other surveys such as living conditions surveys. Could the authors explain what kind of additional information their survey provides compared to official statistics in Singapore?

16) On page 13, second paragraph, the authors say “Waves 3 and 4 included retrospective questions which collected information from periods before and shortly after the lockdown”. However, looking at Table 1, I wonder if it could be a typo since Table 1 says that information in the Wave 3 is retrospective. Also, Table 1 states that the data for May 2020 is "actual", while the text states that it is retrospective.

17) Dependent variable. My main concern comes from the dependent variable because it measures the monthly wage and not the hourly wage. Given that women may have decided (voluntarily or not) to reduce their working hours due to the pandemic, it would be rational to observe a decrease in their monthly wages (because they work fewer hours). How have the hours worked by men and women changed after the pandemic in Singapore according to official statistics? I think it would be important to make a differentiated analysis for full-time and part-time workers. I guess that the survey collects information on hours worked (Table A3 in the appendix). Information on hours worked is essential for the analysis carried out by the authors. Additionally, to contextualise the salary it would be important to provide the average salary of men and women in Singapore (according to official statistics).

18) It is not clear what happens as regards employment situation across the 6 waves. Reading the paper, it seems that during the 6 waves all the people surveyed remain employed, but this would be surprising, even in times of economic expansion there are always flows in the labour market which mean that in certain periods people are not employed. How is it possible to collect monthly wages for 6 periods for married women and their husbands? Are both of them employed during all 6 periods?

19) Classification of teleworkers. The authors define a teleworker as a worker who works fully from home. Because of this definition, the percentage of teleworkers before the lockdown is so low (close to zero in 2018, Figure A1). In my opinion, this is a very rigid definition. Do the results hold when partially relaxing the definition of teleworkers (including at least those who report working mostly from home)?

20) The authors say that during the pandemic, there were people who were able to hold multiple jobs. What percentage of people in the survey are affected by this situation? What is the incidence of moonlighting in Singapore according to official statistics?

21) The essential or non-essential nature of the industry where workers are employed is a very relevant variable for the study of the economic consequences of the pandemic on the labour market. However, the database does not include information on industries. I guess that this is because the questionnaire was originally designed for other purposes before the pandemic (in 2018). While the authors recognize this lack of information, they should put more effort to explain why they consider that not having information on industry would not affect their estimates. Is there any paper specifically addressing the case of Singapore that supports this statement?

22) The authors classify occupations as “professionals” and "non-professional”. I guess that this classification is made to differentiate between teleworkable and non-teleworkable occupations. In my opinion, technicians are also teleworkable occupations.

23) Analysed sample: The analysed sample is unclear. The authors say that their final sample consists of 4638 observations, involving 781 people (362 women and 419 men). I find the description of the analysed sample very confusing as on page 12 they say that the potential participants are 3038, of which 657 people finally met the criteria to participate. It is understood that the people who finally fill in all the waves are 378 (which I understand are women and they report information about their male spouses). If women report information about men, how is it possible that there is more information from men (419) than from women (362)?

24) Since this is an ad-hoc survey, it is very relevant to present information on whether or not the sample is representative of the total population. This is done in Table A2 (Comparison of Baseline Characteristics in the Sample and National Statistics of Married Resident Mothers Aged 30-34). However, it is striking that it compares only its sample of 25-59 year olds with the population aged 30-34. Such a table should compare similar age groups.

25) Table 2. It would be necessary to show the data displayed in Table 2 for the 3 periods. It is also necessary to include the number of hours worked in each of the periods.

RESULTS

26) When analysing the results displayed in Table 4 (DID model estimates) it would be useful to start by commenting on the independent coefficients of “working remotely”, “lockdown” and “post-lockdown”, and then comment on the corresponding interactions terms.

27) However, I find the estimation results in Table 4 rather strange. For example, the coefficient of the “lockdown” variable is 189.070 for the male sample, 116.632 for the female sample, but the corresponding coefficient for the pooled sample is -50.246. How can the authors explain this difference?

28) I also find striking the coefficient of "working remoteley". Figure 2 implies that the monthly wage of teleworkers is higher than the wage of non-teleworkers (before and after the pandemic). However, in the estimates in Table 2, the coefficient of "working remotely" (which measures the difference between teleworkers' and non-teleworkers' wages in the pre-pandemic period), is negative for both men and women. I understand that once the other explanatory factors are controlled for, the sign may change, but this is something the authors should explain adequately in the manuscript.

29) The authors say that “In our total sample (column 1), we see that fully remote work arrangements under lockdown is associated with an increase in monthly earned income by $245.60. The income further increases by $397.02 even after the lockdown is lifted (i.e., November 2020).” What is the theoretical explanation that causes this wage increase? I guess that this finding is related to the number of working hours, but the authors do not explain this result.

30) On the other hand, the effect of the lockdown for teleworkers is the sum of the “lockdown” coefficient plus the interaction term with teleworking (-50 + 245) while for non-teleworkers the effect of the lockdown is just the “lockdown” coefficient (-50). Thus, the pandemic has caused an increase of teleworkers’ wages and a decrease for the non-teleworkers. Again, I think this result is related to the number of working hours. Teleworkers were able to work during the lockdown and non-teleworkers probably reduced the number of working hours (or even do not worked at all) due to the mandatory closures.

31) The authors claim “Time fixed effects are in line with macroeconomic data showing a large dip in economic activity during the lockdown before recovering in November 2020”. However, this statement is not seen in Table 4. For the pooled sample, the “lockdown” coefficient is negative, which would be in line with the previous statement, but for the separate samples, the coefficient of “lockdown” indicates the opposite, as I have mentioned above.

32) Since it is necessary to revise estimations corresponding to the results presented in Table 4, it would also be necessary to revise the estimates presented in Tables 5 to 7.

MINOR COMMENTS:

- Homegeneise the expression Covid19. In some cases it appears as COVID-19 and in others as Covid19.

- Page 17. It says Figure 1, it should be Figure A1 (online Appendix).

6. PLOS authors have the option to publish the peer review history of their article (what does this mean? ). If published, this will include your full peer review and any attached files.

**Do you want your identity to be public for this peer review?** For information about this choice, including consent withdrawal, please see our Privacy Policy .

Reviewer #1: No

Reviewer #2: No

---

## [Author Response · Author response to Decision Letter 0]

30 Apr 2024

Please see the attached file for our responses to reviewers. Thank you.

---

## [Decision Letter · Decision Letter 1]

9 Jun 2024

PONE-D-24-02155R1Unequal Gains from Remote Work during COVID-19 between Spouses: Evidence from Longitudinal Data in SingaporePLOS ONE

Dear Dr. Lee,

Thank you for submitting your manuscript to PLOS ONE. After careful consideration, we feel that it has merit but does not fully meet PLOS ONE’s publication criteria as it currently stands. Therefore, we invite you to submit a revised version of the manuscript that addresses the points raised during the review process.

We look forward to receiving your revised manuscript.

Kind regards,

José Alberto Molina

Academic Editor

PLOS ONE

Reviewers' comments:

Reviewer's Responses to Questions

**Comments to the Author**

1. If the authors have adequately addressed your comments raised in a previous round of review and you feel that this manuscript is now acceptable for publication, you may indicate that here to bypass the “Comments to the Author” section, enter your conflict of interest statement in the “Confidential to Editor” section, and submit your "Accept" recommendation.

Reviewer #1: (No Response)

Reviewer #2: All comments have been addressed

2. Is the manuscript technically sound, and do the data support the conclusions?

Reviewer #1: Partly

Reviewer #2: Partly

3. Has the statistical analysis been performed appropriately and rigorously? 

Reviewer #1: Yes

Reviewer #2: N/A

4. Have the authors made all data underlying the findings in their manuscript fully available?

Reviewer #1: No

Reviewer #2: No

5. Is the manuscript presented in an intelligible fashion and written in standard English?

Reviewer #1: Yes

Reviewer #2: Yes

6. Review Comments to the Author

Reviewer #1: The authors properly addressed the points I previously mentioned in most cases, and the paper becomes much clearer indeed. I have two additional comments and clarifications that come from graphs the authors added or modified.

- It is quite interesting that female teleworkers reduce their hours worked although their hourly wage increases. Then, the natural question is why they do so. I imagined that it was because of more housework or childcare burdens. However, based on Tables S9-1 and S9-2, neither of them increases that much. For example, focus on the coefficient on “Working Remotely x Post-lockdown.” Table 4 says -49.34. First of all, is this -49.34 hours per month? That’s too large! They must increase some other activities because they have a lot more time available. However, Tables S9-1 and S9-2 say that they don’t really increase childcare time or time spent on household chores. We could say the latter increases a bit for some groups, but definitely not enough to spend 49.34 more hours per month. On which activity do they spend time? Is there any way to investigate this more using time use data you might have in your data? Otherwise, the authors’ argument “find that chore responsibilities imposed a negative effect on the female remote workers’ income” may not sound the first order importance given the huge size of the effect. Note that the same thing is true for the group “Working remotely (Fully from home)” even though the average results are driven by this group.

- The same effect size issues are applied for the results – in Table 4, how can male teleworkers' wage increase by $20/hour given their pre-lockdown mean is $30 from Table 2? I understand that it is not fair to just say that the effect size is too large to be trusted because it might be true, but I encourage authors to describe how hourly wages change in a firm in Singapore in general. Is it realistic to have a trend in wages in Figure 2 in terms of the magnitude? For example, some news articles reporting a huge increase in wages in some specific firms in post-COVID period would help readers believe these results. There must be some news or anecdotal evidence if the effect is actually such a big size.

Reviewer #2: See attached file

The manuscript has significantly improved from the previous version. I thank the authors for their detailed responses to my comments and suggestions.

Nonetheless, I consider that the paper requires further revision in several aspects.

Main comments

1) My first comment is regarding the explanatory variables included in the estimations. In particular, the human capital of individuals (measured by their educational level) is an essential determinant of wages and labour supply. Hence, all estimates should include individuals’ education among the explanatories. But the authors claim that the vector Xit only includes age groups, number of children, and presence of an infant under two years old. Additionally, the estimates for the whole sample should include among explanatories a dummy of gender.

2) When analysing their results, the authors make a lot of comparisons between the results for the female and the male subsamples. Nonetheless, such comparisons are simply based on the coefficients (and their significance level), but they are not supported by any statistical test for the equality of regression coefficients.

3) Authors should homogenize the denomination of the analysed indicators through the manuscript. Many times, when the authors refer to “hourly wages” they simply say “wages”. In addition, is it important to correct the inconsistencies along the manuscript as regards the analysed indicators, because in some parts of the manuscript they only refer to “monthly income” (which was the only indicator analysed in the first version of the manuscript), and the new parts of the manuscript refer to the three indicators analysed in the revised manuscript. For instance, on page 33 the authors say “In a sensitivity check, we consider whether monthly income increased monotonically with the extent of remote work arrangements individuals adopted”). In any case, I consider more appropriate the denomination of “monthly wage” than “monthly income”.

4) I am afraid that Table 4 contains a typo, as the coefficients and standard errors of the variables “lockdown” and “post-lockdown” are exactly the same (whereas in Table 5 the coefficients of these variables are different). Consequently, the analysis in the text referring to these coefficients should also be corrected.

5) In Table 6 there are also results that seem somewhat strange. It could may be a typo?. The coefficients of the interaction term between “remote”, “lockdown” and “receiving any help” (panel C, Table 6) and the corresponding to the interaction term between “remote”, “lockdown” and “spouse working remotely” (panel D) are 0.01 with a standard error of 0.00 in all estimates for panel C and D.

6) Authors claim that they estimate the causal effects of the adoption of remote work arrangements on wages and hours worked as a result of the pandemic. In my opinion this is not true since the authors select a sample of individuals that were employed across six points in time (waves) and compare the results for observations of “fully working from home” and “non fully-working from home”. Particularly, the authors say that “The nation-wide lockdown mandate, observed from 7 April 2020 to 1 June 2020, required workers whose occupational tasks could be carried out remotely to work from home regardless of personal preferences [28]. The mandate was strictly enforced, thereby alleviating issues of self-selection into remote work arrangements”. The mandatory fact of the adoption of telework would allow to analyse the causal effect of telework by defining a treatment group (those who were not working from home before the lockdown but working from home during the lockdown) and two potential control groups: 1) those who were working from home before the lockdown and also during the lockdown (because for them working from home is the result of their preferences), and 2) those who were not working from home before the pandemic nor during the lockdown. If the authors maintain their empirical strategy of comparing fully teleworkers with non-fully teleworkers, I consider they should not claim to be analysing the causal effect.

7) On page 17, the authors claim, “As our data initially undersampled employed persons, our sample selection strategy, in a way, helped bring our sample’s employment rates to be closer those reported in the official statistics”. I disagree with this statement because the authors are introducing a significant selection bias by excluding from the analysis all persons who have not been employed at some point in time.

8) Table 2. It seems quite strange that the percentage of teleworkers during the period of lockdown is lower than that in the pre-pandemic period (56.12%, 51.64% and 18.67% among females; the male sample display a similar path). In my opinion, the explanation for this phenomenon is somewhat superficial and not supported by their data (“…possibly due to job switches or changes in the number of jobs held.”). They should explore further to give a substantiated explanation. I am afraid it is due to the sample selection. I would expect an increase in the percentage of teleworkers over the total number of employed people in the May 2020 survey (without excluding any employed individual) compared to the corresponding percentage in the April 2018 survey.

Other comments:

• The objective of the paper described on page 4 (first paragraph) is not clear. I suggest reviewing its redaction.

• As I suggested in my first report, the authors have included the percentage of workers working from home during the pandemic. Still, they do not report the information on the corresponding figures before the pandemic (page 3). It would be convenient to report it. In Western countries, the percentage of workers working from home has increased significantly compared with the figures before the pandemic. I guess that something similar could have happened in Singapore.

• Database description:

The database description is more precise in this version of the manuscript.

Perhaps the description could be even clearer by explaining that the final sample consists of 335 females and 384 males who were employed during the 6 waves of the survey, which implies 2007 observations for women and 2301 observations for men.

When describing the database and the timing of the data collection (pages 14-15), the authors state that there were three surveys: April 2018, May 2020, and November 2020. However, on page 14, they claim they added questions about COVID-19 at the end of 2019. If I have understood correctly, they added the questions in the May 2020 survey collecting retrospective information about December 2019.

The manuscript also contains many inconsistencies regarding the time of the survey. It is important to homogenise the survey periods. For instance, in some parts of the manuscript, they say that the first survey was collected in April 2018, but in other parts, they say April-July 2018 (even in the axis of Figure 2).

It is clear in the Conclusions section that individuals surveyed are all females and that they report the information on their male spouses. I suggest indicating this in the main text (maybe on page 22 when referring to the participants in the survey).

• Page 9: When referring to the work of Atkin et al., it would also be helpful to indicate based on which data (country) the work of Atkin et al. was carried out (similar to what is done when referring to the work of Gibbs et al.).

• Page 21: The authors say that in their sample, the mean value of the number of children is 1.5 children. I suggest indicating its distribution instead of the mean value (for instance: no children, one child, two children, three or more). Despite this, it is unclear if all females in the sample have children because the authors do not mention this feature in the sample selection description, but they say, “Using an individual-level panel dataset of married spouses with children” in the Discussion section. The authors should clarify this.

• Table 2: The dependent variables (hourly wages, monthly earnings and monthly hours worked) should appear disaggregated for teleworkers and non-teleworkers.

It would also be helpful to display the distribution of workers following the classification used in the sensitivity analysis (whether working fully from home, mostly from home, half from home and half outside the home, mostly outside the home or fully outside the home.)

On the other hand, since the persons surveyed along the six waves are the same, it is strange that the nationality of persons surveyed in the post-lockdown period are somewhat different. Similarly, it is very strange that the percentage of women with children under two years is 1% in the pre-pandemic period but increases up to 40% in the lockdown period. It seems that the question in the questionary of April 2018 was different than that in May 2020 and November 2020.

• The analysis of the results should be more thorough. For instance, the authors claim, “The estimated coefficients of the standalone “remote work” variable suggest that the WFH arrangements lowered income for both genders (a decrease of $393.37 for men and $53.75 for women) even though the coefficient is significant only among men (columns 4-6).” These coefficients do not indicate a drop; they indicate that before the pandemic, teleworkers earned lower wages than non-teleworkers. Other sentences display similar inconsistencies.

Minor comments

• Page 7: The authors say, “Employment rate also turned positive in the last quarter of 2020.” Nevertheless, the data in Figure 1 seems negative in 2020.Q4, and it turns positive in the first quarter of 2021. There is also a typo; they should say “employment growth” (or “employment growth rate”) instead of “employment rate” (the employment rate is the percentage of employed persons over the total population aged over 16 years old).

• It is necessary to homogenise the vertical and horizontal axis in Figure 1 for GDP growth and employment growth.

• Page 11: The figures in the first sentence are not clear.

• Table 1: The note 2 under Table 1 says “across four points in time”. I understand that it should be “across three points in time”.

• Page 12 and 17: The expression “during this period” is duplicated in the sentence “During this period, schools and childcare centers were closed during this period as the law mandated ……. “ (page 12). The sentence “Further excluded were 360 observations (all observations for 60 individuals) for having worked in waves 1-2 but stopped working in waves in 2020” is duplicated. (page 17)

• Page 30: The title of section 5.2 should be actualized.

7. PLOS authors have the option to publish the peer review history of their article (what does this mean? ). If published, this will include your full peer review and any attached files.

**Do you want your identity to be public for this peer review?** For information about this choice, including consent withdrawal, please see our Privacy Policy .

Reviewer #1: No

Reviewer #2: No

---

## [Author Response · Author response to Decision Letter 1]

26 Aug 2024

Please see attached document for response to reviewers. Thank you.

---

## [Decision Letter · Decision Letter 2]

1 Nov 2024

PONE-D-24-02155R2Unequal Gains from Remote Work during COVID-19 between Spouses: Evidence from Longitudinal Data in SingaporePLOS ONE

Dear Dr. Lee,

Thank you for submitting your manuscript to PLOS ONE. After careful consideration, we feel that it has merit but does not fully meet PLOS ONE’s publication criteria as it currently stands. Therefore, we invite you to submit a revised version of the manuscript that addresses the points raised during the review process.

We look forward to receiving your revised manuscript.

Kind regards,

José Alberto Molina

Academic Editor

PLOS ONE

Reviewers' comments:

Reviewer's Responses to Questions

**Comments to the Author**

1. If the authors have adequately addressed your comments raised in a previous round of review and you feel that this manuscript is now acceptable for publication, you may indicate that here to bypass the “Comments to the Author” section, enter your conflict of interest statement in the “Confidential to Editor” section, and submit your "Accept" recommendation.

Reviewer #1: All comments have been addressed

Reviewer #2: (No Response)

2. Is the manuscript technically sound, and do the data support the conclusions?

Reviewer #1: (No Response)

Reviewer #2: Partly

3. Has the statistical analysis been performed appropriately and rigorously? 

Reviewer #1: (No Response)

Reviewer #2: N/A

4. Have the authors made all data underlying the findings in their manuscript fully available?

Reviewer #1: (No Response)

Reviewer #2: No

5. Is the manuscript presented in an intelligible fashion and written in standard English?

Reviewer #1: (No Response)

Reviewer #2: Yes

6. Review Comments to the Author

Reviewer #1: (No Response)

Reviewer #2: See attached file

The manuscript has improved from the previous version, and the authors have addressed the majority of my comments and suggestions. I thank the authors for their detailed responses. I have some additional comments, clarications, and corrections of several inconsistencies that require further revision.

Table 2:

As I suggested in my previous report, the most relevant information for the paper's objectives should appear in the main text. In particular, part of the information that is now displayed in the new Table S10 and Table S9 should be included in Table 2. I agree that Table 2 is too long, but in my opinion the numbers in parenthesis (are they SD? it is not indicated what they are) could be eliminated. It is not usually to display the SD for frequencies. Logically, this comment does not apply to the endogenous variables. Hence, the endogenous variables should appear disaggregated for teleworkers and non-teleworkers (the rst 6 rows from Table S10). Also, the time dedicated to childcare and chores should appear disaggregated for teleworkers and non- teleworkers.

The explanation oered by the authors as regards the decrease in the percentage of teleworkers in the lockdown period and the post-lockdown period is not convincing. There is not only a switch from “Working Fully from Home” to “Working Mostly from Home”, but the most preoccupant part is that their data show an increase in the percentage of workers who declare “working Fully outside” or “mostly outside” (from 32.81% in the pre-lockdown period to 34.72% in lockdown and 52.23% in post-lockdown among males). Something similar occurs for females since the percentage of women who declare “working fully outside” or “mostly outside” increases from 16.41% in the pre-lockdown period to 18.91% in the lockdown and 34.04% in the post-lockdown period. This characteristic of their sample selection should be indicated as an additional limitation of their conclusions. Additionally, the authors start their motivation in the abstract with the sentence, “The COVID-19 outbreak and the rise of remote work generated distinct labor market outcomes for workers based on their occupation and the ability to adapt to changes”, when their data show precisely the contrary pattern.

The decrease in the percentage of people working from home is coherent with the occupational changes observed during the lockdown and the post-lockdown periods. In their data, the percentage of professionals and associates professionals (the most teleworkable occupations according to the international empirical evidence) decreases during the lockdown and the post- lockdown period. In contrast, the percentage of service workers clearly increases. Once more, this characteristic of their sample selection is contrary to the empirical evidence of the pandemic on the labour market: the pandemic provoked employment losses in non-teleworkable occupations. Could it be a problem of attrition in their database? The authors claim that they exclude individuals who were not employed in any of the 6 waves from the panel. Does the same apply to “missing individuals”? In that case, it could be feasible that some individuals who have been excluded from the panel are employed individuals (maybe teleworkers) who were not available to respond to the survey.

I miss a brief explanation (similar to what the authors explained in the responses to my previous report) regarding the huge increase in the percentage of women having an infant aged two years or younger (from 1% to 40% during lockdown/post-lockdown).

On page 24, the authors claim that “For instance, while similar shares of men and women were in professional (about 80%) and associate professional (10%) jobs, more women were in clerical jobs (8% as opposed to 2% among men) which could be easily adopt full remote work arrangements.” This statement is not true since the international literature has shown that professionals and associate professionals are more teleworkable than clerical workers.

As regards the comments in Table 2, it should be noted that the gender gap in the time devoted to childcare has been reduced during the lockdown and the post-lockdown, while the gender gap in the time devoted to chores has increased.

Finally, I consider that Table 2 should contain tests of dierences in means between males and females.

Explanatory variables:

Page 27: After reading the explanation of the independent variables included in the vector Xit, it seems that it includes industries. However, in the conclusions section, the authors clarify that they do not have information on industry. This aspect should also be claried in the description of the Xit vector.

Results:

There are some inconsistencies in the interpretation of the results. As I indicated in my second report, the interpretation of the “Remote work” coecient in Table 4 is not correct. Since the reference category in estimates is the pre-lockdown period, the coecient of “remote work” measures the dierence in hourly wages, monthly income, and monthly hours worked between teleworkers and non-teleworkers in the pre-lockdown period. Please revise the comments on this coecient.

The interpretation of the coecients of the interaction terms (third paragraph on page 33) seems somewhat confusing. In my opinion, it would be more precise to start the explanation with the authors' comments in the second paragraph on page 34, where they correctly interpret the magnitude of the eect of the lockdown (and the post-lockdown) on teleworkers and non-teleworkers. Attending to the authors' correct statement on page 34, during the lockdown period, male WFH workers experience income gains of $204.93 (and $596.46 during post- lockdown), while non-WFH workers experience income losses by -$182.50 (and income gains by $150.04 in the post-lockdown). The interaction term "Working remotely x Lockdown" allows to test the null hypothesis that the eect of the lockdown does not depend on the type of workers (teleworkers vs. non-teleworkers). The coecient of the interaction term tests whether the lockdown has had a dierential eect on teleworkers and non-teleworkers. Hence, the value of the interaction term ($387.43 for men) measures the (statistically signicant) dierential eect of the lockdown on the monthly wages between male teleworkers and non-teleworkers ($204.93 - (-$182.50)=$387.43). Comments on other interaction terms and those in Table 5 have to be revised in the same line.

On page 38, the authors claim, “This was consistent with the pattern seen in Table 4, once again suggesting the sub-par nature of jobs that were done from home before COVID-19 and the global popularization of remote work”. This explanation should be indicated when analysing the results in Table 4.

Regarding Table 6, the authors indicate that “The standalone components of the 3-way interaction were included in the estimations but were not shown”. The table, including the 3- way interactions, should be displayed as a supplementary table.

On page 38, the authors highlighted, “Interestingly, among women, those working mostly outside also experienced a rise in monthly income during the lockdown.” Nevertheless, the coecient is not statistically signicant, so they cannot conclude the above statement.

On page 44, in the discussion section, it is not clear where the gures for “the increased monthly income generated by adopting WFH arrangements under lockdown/post-lockdown constitutes 8.2-9.5% growth for men while comprising mere 1.2-1.5% (statistically insignicant) growths for women” come from.

On page 46, the authors state that “Despite the above-mentioned dierences across gender, the mediator analyses lends support for Hypothesis 3 regarding the role of chore burdens, availability of external help for housework, and availability of WFH spouses on dictating the extent of income and productivity(wage) gains of the remotely working individuals.” It would also be convenient to indicate that they do not nd evidence regarding the role of childcare.

Among the study's limitations, it should also be recognised that the sample is underrepresented in terms of employment.

Other comments:

- It is necessary to homogenise the monetary units across the paper, which appear as SGD, S$, or $.

- On page 22, I guess there is a typo: substitute “they were fully WFH: For both genders,” with “they were fully WFH. For both genders,”

- Review the web links in the references list. For instance, the link https://www.ilo.org/resource/brief/working-home-estimating-worldwide-potential does not exist any more.

- The authors refer to references 8 and 10 on page 3. I am afraid that the reference 10 (A D Michael;Groen,Jerey. Bureau of Labor Statistics. [cited 2024 Apr 26]. Telework during the COVID-19 pandemic: estimates using the 2021 Business Response Survey : Monthly Labor Review: U.S. Bureau of Labor Statistics) does not say nothing about Singapore.

- The wording of the last sentence in the rst paragraph on page 7 is confusing. I guess there is a typo, and I wonder if it should be two sentences instead of one.

- The dates of the second interview on page 15 (second paragraph) are inconsistent: May 2020 vs. May-June 2020.

- Page 17: Substitute “61.44% of the total of 3,960 females” for “61.44% of the total of 3,960 female person-wave observations”.

- It should be indicated under Tables which statistical test is used to contrast the equality of the coecients of the male and female subsamples.

- On page 41, indicate panel A, panel B, and panel C when commenting on the results from Table 6 (the word “panel” is missing in some of the parenthesis).

7. PLOS authors have the option to publish the peer review history of their article (what does this mean? ). If published, this will include your full peer review and any attached files.

**Do you want your identity to be public for this peer review?** For information about this choice, including consent withdrawal, please see our Privacy Policy .

Reviewer #1: No

Reviewer #2: No

---

## [Author Response · Author response to Decision Letter 2]

7 Jan 2025

Please see attached document for our response to reviewers. Thank you.

---

## [Decision Letter · Decision Letter 3]

21 Feb 2025

PONE-D-24-02155R3Unequal Gains from Remote Work during COVID-19 between Spouses: Evidence from Longitudinal Data in SingaporePLOS ONE

Dear Dr. Lee,

Thank you for submitting your manuscript to PLOS ONE. After careful consideration, we feel that it has merit but does not fully meet PLOS ONE’s publication criteria as it currently stands. Therefore, we invite you to submit a revised version of the manuscript that addresses the points raised during the review process.

We look forward to receiving your revised manuscript.

Kind regards,

José Alberto Molina

Academic Editor

PLOS ONE

Journal Requirements:

Reviewers' comments:

Reviewer's Responses to Questions

**Comments to the Author**

1. If the authors have adequately addressed your comments raised in a previous round of review and you feel that this manuscript is now acceptable for publication, you may indicate that here to bypass the “Comments to the Author” section, enter your conflict of interest statement in the “Confidential to Editor” section, and submit your "Accept" recommendation.

Reviewer #2: All comments have been addressed

2. Is the manuscript technically sound, and do the data support the conclusions?

Reviewer #2: Partly

3. Has the statistical analysis been performed appropriately and rigorously? 

Reviewer #2: N/A

4. Have the authors made all data underlying the findings in their manuscript fully available?

Reviewer #2: No

5. Is the manuscript presented in an intelligible fashion and written in standard English?

Reviewer #2: Yes

6. Review Comments to the Author

Reviewer #2: See attached file

The manuscript has improved from the previous version, and the authors have addressed the majority of my comments and suggestions. I thank the authors for their detailed responses. I have some additional comments, clarifications, and corrections of several inconsistencies that require further revision.

Main comments:

1) In the descriptive data in Table 2, the authors should comment that in pre-lockdown, lockdown and post-lockdown, WFH workers earn more than non-WFH workers (both in terms of monthly salary and hourly wage). This analysis is especially relevant because the estimates in Table 4 show the opposite result. According to the estimates in Table 4, before the lockdown period WFH workers earned less than non-WFH workers. Authors should provide an explanation for this contrary result between the descriptive analysis and their estimates.

2) Due to the above comment, I am concerned about the econometric model specification (equation 1, page 27) that has significant consequences for interpreting the results. I am afraid there is a misunderstanding in interpreting the coefficients of the “lockdown” and “post-lockdown” variables. Apart from the “lockdown” dummy (equal to 1 for waves 3, 4, and 5) and the “post-lockdown” dummy (equal to 1 for wave 6), the estimates include time-fixed effects, occupation dummies, and the interactions between occupation and time-fixed effects. Regarding time-fixed effects, I guess the authors have included 5 dummies for the 6 waves, and the reference category is wave 1 (April-July 2018). If this is the case, the “lockdown” coefficient would measure the differential only against the first wave and not against the entire pre-lockdown period, as the authors point out. But additionally, since the “lockdown” dummy includes 3 waves (3, 4, and 5), the coefficient of “lockdown” measures the difference between wave 4 and wave 1 (because part of the effect of the lockdown is captured by the time-fixed effects that are not displayed in Table 4). Furthermore, the model includes interactions between occupation (4 categories?) and time-fixed effects (they do not indicate which category is the reference in estimates; I guess that is the first one, “Professionals”). These interaction terms make the interpretation of the coefficients even more complicated, since now, the reference category of the occupation also conditions the reference category for the period. The authors must clarify this point. If my interpretation were proper, all comments should be reviewed.

3) As I indicated in my previous report, it is also important that authors carefully review the interpretation of the interaction terms as there continue to be inconsistencies in their comments. I point out a single example, but it is necessary to review all of them. Page 46: “In the post-lockdown period, female WFH workers’ monthly hours dropped by as much as 49.34 hours/month”. According to estimates showed in Table 4, in the post-lockdown period, female WFH workers’ monthly hours dropped by as much as 36.49 hours/month, and an increase in hours work of 12.85 for female non-WFH. Hence, in the post-lockdown period the difference between WFH workers y non-WFH workers is -49.34 hours/month. Logically, this interpretation would also need to change if my second comment were right.

4) Page 43: The authors compare their estimated coefficients with the official mean wages for men and women in Singapore (SGD $4,719 and SGD $4,374, respectively). This is not correct, their coefficients are only comparable with the average values of their data, which, as the authors themselves point out, are not comparable with the total universe because their data is a subsample of married men and women.

Other comments:

5) Page 3. Indicate the source for the reported data for Netherlands (47.4%), Denmark (53.0%), and U.S. (50%).

6) Page 4. Substitute “mediating roles of housework” by “mediating roles of household responsibilities” in the sentence “Lastly, we explore potential explanations for these gender differences by exploring the mediating roles of housework.”

7) Page 7. Substitute “the rate of deceleration of GPD slowed down” by “the rate of decline of GPD slowed down”.

8) Page 7, Figure 1. Labels in horizontal axis in panel A and panel B should be homogenised. In addition, the numbers in vertical axis appear with % in panel A but not in panel B.

9) Page 15, first paragraph. Delete “hourly wages” in “(monthly salary income, monthly work hours, hourly wages)” (the questionary does not ask about hourly wages).

10) Page 16. The authors claim that they interviewed 660 persons, and then they dropped 5 persons that were not married. Then, they drop those not employed persons across the six waves. If the number of women in the final sample is 335, this implies that they dropped 320 women (49% of 655). The final sample for men is 384, meaning they dropped 271 men (41% of 655). Nevertheless, in the Conclusions section, the authors claim that “the number of respondents we dropped from our sample due to unemployment in 2020 comprised only 8.3% (360 person-wave observations across six survey waves, or 60 unique individuals)of the total sample of 4,308 person-wave observations.”

11) Page 27, equation 1. The variables Lockdown and Postlockdown should not include temporal subscripts.

12) Page 25. Typo. “another: A plurality” should be “another. A plurality”.

13) Page 52. Typo: Include a space in “60 unique individuals)of”

7. PLOS authors have the option to publish the peer review history of their article (what does this mean? ). If published, this will include your full peer review and any attached files.

**Do you want your identity to be public for this peer review?** For information about this choice, including consent withdrawal, please see our Privacy Policy .

Reviewer #2: No

---

## [Author Response · Author response to Decision Letter 3]

4 Apr 2025

Please see attached file with our responses to reviewer comments. Thank you.

---

## [Decision Letter · Decision Letter 4]

22 Apr 2025

Unequal Gains from Remote Work during COVID-19 between Spouses: Evidence from Longitudinal Data in Singapore

PONE-D-24-02155R4

Dear Dr. Lee,

We’re pleased to inform you that your manuscript has been judged scientifically suitable for publication and will be formally accepted for publication once it meets all outstanding technical requirements.

Kind regards,

José Alberto Molina

Academic Editor

PLOS ONE

Additional Editor Comments (optional):

Reviewers' comments:

Reviewer's Responses to Questions

**Comments to the Author**

1. If the authors have adequately addressed your comments raised in a previous round of review and you feel that this manuscript is now acceptable for publication, you may indicate that here to bypass the “Comments to the Author” section, enter your conflict of interest statement in the “Confidential to Editor” section, and submit your "Accept" recommendation.

Reviewer #2: All comments have been addressed

2. Is the manuscript technically sound, and do the data support the conclusions?

Reviewer #2: (No Response)

3. Has the statistical analysis been performed appropriately and rigorously? 

Reviewer #2: Yes

4. Have the authors made all data underlying the findings in their manuscript fully available?

Reviewer #2: No

5. Is the manuscript presented in an intelligible fashion and written in standard English?

Reviewer #2: Yes

6. Review Comments to the Author

Reviewer #2: The authors have done a good job in incorporating my comments and suggestions. I thank the authors for their detailed responses. I have no further comments. Thank you for your review.

7. PLOS authors have the option to publish the peer review history of their article (what does this mean? ). If published, this will include your full peer review and any attached files.

**Do you want your identity to be public for this peer review?** For information about this choice, including consent withdrawal, please see our Privacy Policy .

Reviewer #2: No

---

## [Editor Report · Acceptance letter]

PONE-D-24-02155R4

PLOS ONE

Dear Dr. Lee,

I'm pleased to inform you that your manuscript has been deemed suitable for publication in PLOS ONE. Congratulations! Your manuscript is now being handed over to our production team.

Kind regards,

on behalf of

Professor José Alberto Molina

Academic Editor

PLOS ONE